# The functional O-mannose glycan on α-dystroglycan contains a phospho-ribitol primed for matriglycan addition

Jeremy L Praissman[1†], Tobias Willer[2,3,4,5†], M Osman Sheikh[1†], Ants Toi[6], David Chitayat[7,8,9], Yung-Yao Lin[10,11,12], Hane Lee[13,14,15], Stephanie H Stalnaker[1], Shuo Wang[1], Pradeep Kumar Prabhakar[1], Stanley F Nelson[13,14,15], Derek L Stemple[12], Steven A Moore[16], Kelley W Moremen[1,17], Kevin P Campbell[2,3,4,5]*, Lance Wells[1,17]*

[1]Complex Carbohydrate Research Center, University of Georgia, Athens, United States; [2]Department of Molecular Physiology and Biophysics, Carver College of Medicine, University of Iowa, Iowa City, United States; [3]Howard Hughes Medical Institute, University of Iowa, Iowa City, United States; [4]Department of Neurology, Carver College of Medicine, University of Iowa, Iowa City, United States; [5]Department of Internal Medicine, Carver College of Medicine, University of Iowa, Iowa City, United States; [6]Department of Medical Imaging, Mount Sinai Hospital, Toronto, Canada; [7]Division of Clinical and Metabolic Genetics, The Hospital for Sick Children, University of Toronto, Toronto, Canada; [8]The Prenatal Diagnosis and Medical Genetics Program, Mount Sinai Hospital, Toronto, Canada; [9]Department of Obstetrics and Gynecology, University of Toronto, Toronto, Canada; [10]Blizard Institute, London, United Kingdom; [11]Barts and The London School of Medicine and Dentistry, Queen Mary University of London, London, United Kingdom; [12]Wellcome Trust Genome Campus, Wellcome Trust Sanger Institute, Hinxton, United Kingdom; [13]Department of Human Genetics, University of California, Los Angeles, Los Angeles, United States; [14]David Geffen School of Medicine, University of California, Los Angeles, Los Angeles, United States; [15]Department of Pathology and Laboratory Medicine, University of California, Los Angeles, Los Angeles, United States; [16]Department of Pathology, Carver College of Medicine, University of Iowa, Iowa City, United States; [17]Department of Biochemistry and Molecular Biology, University of Georgia, Athens, United States

*For correspondence: kevin-campbell@uiowa.edu (KPC); lwells@ccrc.uga.edu (LW)

[†]These authors contributed equally to this work

Competing interests: The authors declare that no competing interests exist.

**Abstract** Multiple glycosyltransferases are essential for the proper modification of alpha-dystroglycan, as mutations in the encoding genes cause congenital/limb-girdle muscular dystrophies. Here we elucidate further the structure of an *O*-mannose-initiated glycan on alpha-dystroglycan that is required to generate its extracellular matrix-binding polysaccharide. This functional glycan contains a novel ribitol structure that links a phosphotrisaccharide to xylose. ISPD is a CDP-ribitol (ribose) pyrophosphorylase that generates the reduced sugar nucleotide for the insertion of ribitol in a phosphodiester linkage to the glycoprotein. TMEM5 is a UDP-xylosyl transferase that elaborates the structure. We demonstrate in a zebrafish model as well as in a human patient that defects in TMEM5 result in muscular dystrophy in combination with abnormal brain development. Thus, we propose a novel structure—a ribitol in a phosphodiester linkage—for

the moiety on which TMEM5, B4GAT1, and LARGE act to generate the functional receptor for ECM proteins having LG domains.

## Introduction

Twenty-five years ago it was proposed that alpha-dystroglycan (α-DG) played a major role in bridging the extracellular matrix to the muscle plasma membrane and actin cytoskeleton as a component of the multi-protein dystrophin glycoprotein complex (*Ervasti et al., 1990*; *Yoshida and Ozawa, 1990*). Defects in the proper formation of this complex have been shown to be causal for various forms of muscular dystrophy (*Carmignac and Durbeej, 2012*). However, to date, only 3 patients have been found that have primary defects in the coding sequence of α-DG (*Dong et al., 2015*; *Geis et al., 2013*; *Hara et al., 2011*). Over fifteen years ago, it was suggested that defects in the glycosyltransferases needed for proper glycosylation of α-DG were causal for a subset of muscular dystrophies, the so-called secondary dystroglycanopathies that can range in severity from mild Limb-Girdle Muscular Dystrophy (LGMD) to severe Walker-Warburg syndrome (WWS) (*Chiba et al., 1997*; *Holt et al., 2000*; *Wells, 2013*). In the early 2000's, it became clear that the defects causal for disease involved enzymes that initiated and elaborated the extended *O*-mannose (*O*-Man) glycan structures covalently attached to Ser/Thr residues of α-DG (*Beltrán-Valero de Bernabé et al., 2002*; *Michele et al., 2002*). Since then, steady progress has been made in elucidating the subset of mammalian *O*-Man structures that directly interact with extracellular matrix components and the candidate genes necessary for the functional glycosylation of α-DG (*Praissman and Wells, 2014*). These include the identification of a subset of phosphorylated *O*-Man structures containing extended, LARGE-dependent, repeating disaccharide polymers, structures that have been recently termed matriglycan (*Yoshida-Moriguchi and Campbell, 2015*).

*O*-Mannosylation begins in the endoplasmic reticulum (recently reviewed in *Dobson et al., 2013*; *Endo, 2015*). The addition of *O*-Man in an alpha linkage to serine and threonine residues of a select set of mammalian glycoproteins is catalyzed by the POMT1/2 complex using dolichol-phosphomannose as the donor (*Beltrán-Valero de Bernabé et al., 2002*). At this point, by mechanisms that have yet to be fully elucidated, there is a divergence in elaboration (*Figure 1*). The vast majority of *O*-Man sites are extended in the Golgi in a beta-1,2 linkage with N-acetylglucosamine (GlcNAc) by POMGNT1 that can then be further branched by GlcNAc and/or elaborated by galactose, fucose, and sialic acid to generate the M1 and M2 glycans (*Praissman and Wells, 2014*; *Lee et al., 2012*; *Yoshida et al., 2001*). A small subset of O-Man modified sites, apparently exclusively on α-DG, are extended in the endoplasmic reticulum by a GlcNAc in a beta-1,4 linkage by POMGNT2 to generate the M3 glycans (*Manzini et al., 2012*; *Yoshida-Moriguchi et al., 2013*). This is further elaborated into a trisaccharide by the action of a beta-1,3-N-acetylgalactosamine (GalNAc) transferase, B3GALNT2 (*Yoshida-Moriguchi et al., 2013*). This trisaccharide is a substrate for POMK that phosphorylates the initiating *O*-Man residue at the 6-position (*Yoshida-Moriguchi et al., 2013*). After unknown elaboration of the phosphotrisaccharide in a phosphodiester linkage, presumably by one or more of the as-yet to be assigned CMD-causing gene products (ISPD, TMEM5, Fukutin, and FKRP), B4GAT1 adds a glucuronic acid (GlcA) in a beta-1,4 linkage to an underlying beta-linked xylose (Xyl) (*Praissman et al., 2014*; *Willer et al., 2014*). The addition of the xylose by an unspecified enzyme followed by the action of B4GAT1 serves as a primer for LARGE to synthesize the repeating Xyl-GlcA disaccharide, matriglycan, that serves as the binding site for several extracellular matrix proteins (*Praissman et al., 2014*; *Willer et al., 2014*).

Here we report a M3 glycan structure with a phosphodiester linked ribitol that TMEM5, B4GAT1, and LARGE act on to generate the functional receptor for extracellular matrix (ECM) ligands characterized by Laminin G domain-like (LG) protein domains. We demonstrate that ISPD is a CDP-ribitol pyrophosphorylase employed for the synthesis of the required sugar (alcohol) nucleotide needed for ribitol insertion into the M3 glycan. We establish TMEM5 as the candidate xylose transferase and demonstrate the impact of its knockdown in a zebrafish model consistent with a CMD phenotype. We also identify and characterize a novel TMEM5 mutation identified in a WWS family. Finally, we present a model of the functional M3 glycan structure along with the more than a dozen assigned or proposed enzymes required for its synthesis.

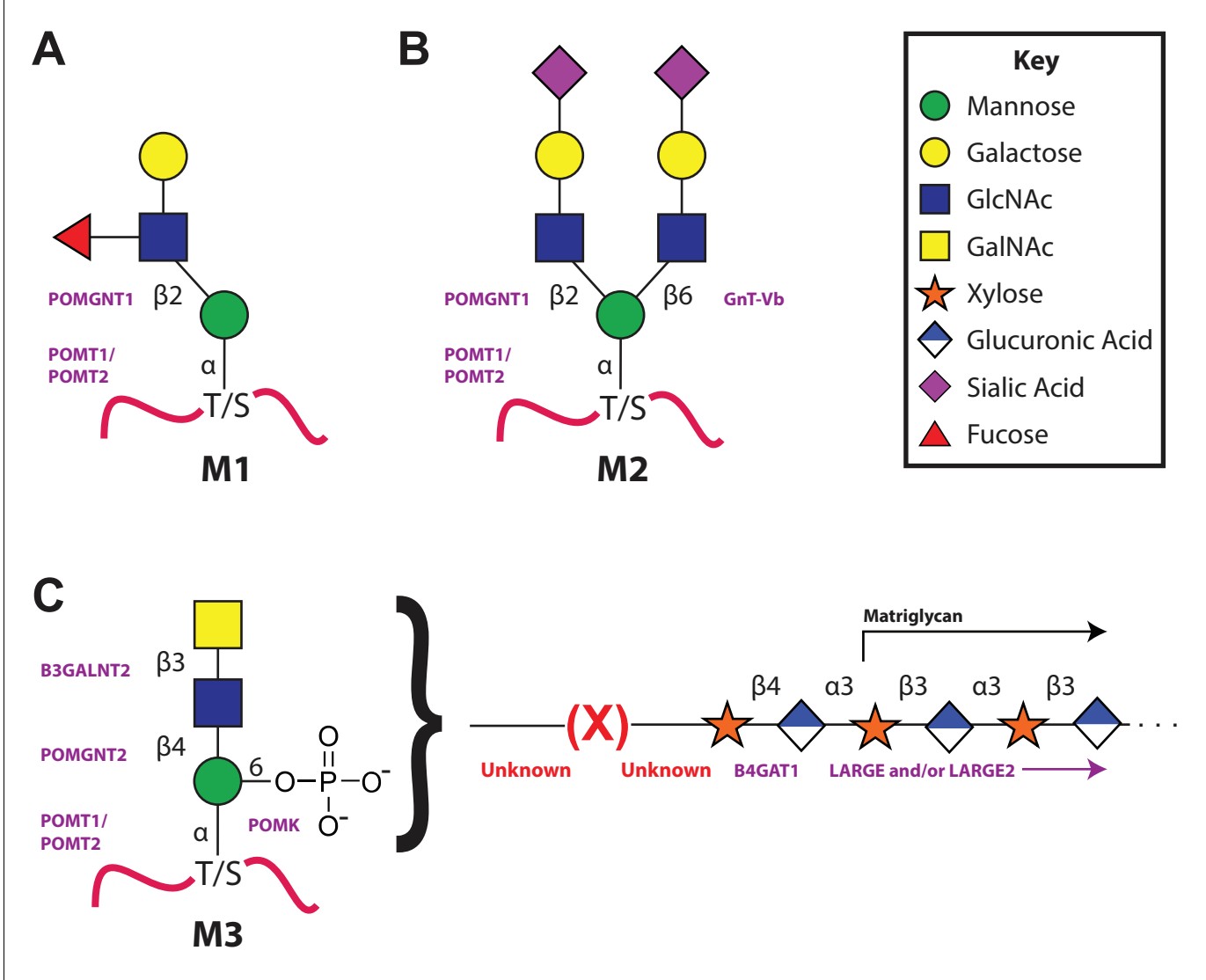

**Figure 1.** The 3 subclasses of the O-Mannosylation pathway. Following addition of Man to serine/threonine residues on protein substrates by POMT1/2 there is a divergence in elaboration. Known enzymes are displayed in purple and unknown enzymes prior to this manuscript are displayed in red. (**A**) POMGnT1 adds a beta-2 linked GlcNAc onto the underlying Man. Shown is just one potential extended M1 glycan structure. (**B**) Following the action of POMGnT1, GnT-Vb can add a beta-6 linked GlcNAc onto a M1 glycan structure to convert it to a M2 glycan structure. Shown is just one potential extended M2 glycan structure. (**C**) Instead of extension in a beta-2 linkage, M3 glycan structures get the core Man extended by POMGNT2 with a beta-4 linked GlcNAc. This disaccharide is further elaborated to contain a phosphodiester linkage to an unknown moiety (shown as (X)) that becomes modified with repeats of (Xyl-GlcA).

## Results

### Ribitol-Xyl-GlcA is released from α-DG-340 following cleavage of the phosphodiester linkage

A truncated, secreted version of α-DG with a COOH-terminal GFP and His tag (α–DG-340 that is only 28 amino acids (313–340) derived from α-DG following endogenous furin cleavage) was overexpressed in HEK293F cells, which express low levels of LARGE, and purified from the media. This 28 amino acid sequence contains only one putative M3 glycan consensus site (TPT, 317–319). We previously demonstrated that the generation of peptides bearing the phosphotrisaccharide could be isolated from α-DG constructs following cleavage of the phoshodiester linkage and that such treatment resulted in loss of IIH6 (an antibody that binds functionally glycosylated α-DG) binding (*Figure 2—*

figure supplement 2) (*Yoshida-Moriguchi et al., 2010*). Thus, we decided to interrogate the released glycan portion to further elucidate the functional glycan structure. Following aqueous HF treatment, which selectively cleaves phosphodiesters, released glycans were isolated from α–DG-340 and permethylated. Tandem mass spectrometry of the permethylated glycans revealed the presence of a linear pentitol-Xyl-GlcA (*Figure 2*) as well as further disaccharide extended structures (pentitol-(Xyl-GlcA)n, *Figure 2—figure supplement 1*).

## ISPD is a CDP-ribitol (ribose) pyrophosphorylase

Given the identification of pentitol in the glycan structure, we investigated whether ISPD might be able to generate CDP-ribitol. Our rationale for this was that mutations in ISPD are causal for CMD (*Roscioli et al., 2012*; *Willer et al., 2012*), ISPD is a putative nucleotidyltransferase found in the cytosol (*Vuillaumier-Barrot et al., 2012*), and homologs in bacterial systems are involved in the generation of identical, CDP-ribitol (*Baur et al., 2009*), or similar structures, 4-CDP-2-C-methyl-D-erythritol (*Richard et al., 2001*). ISPD was expressed as a His fusion protein in HEK293F cells and

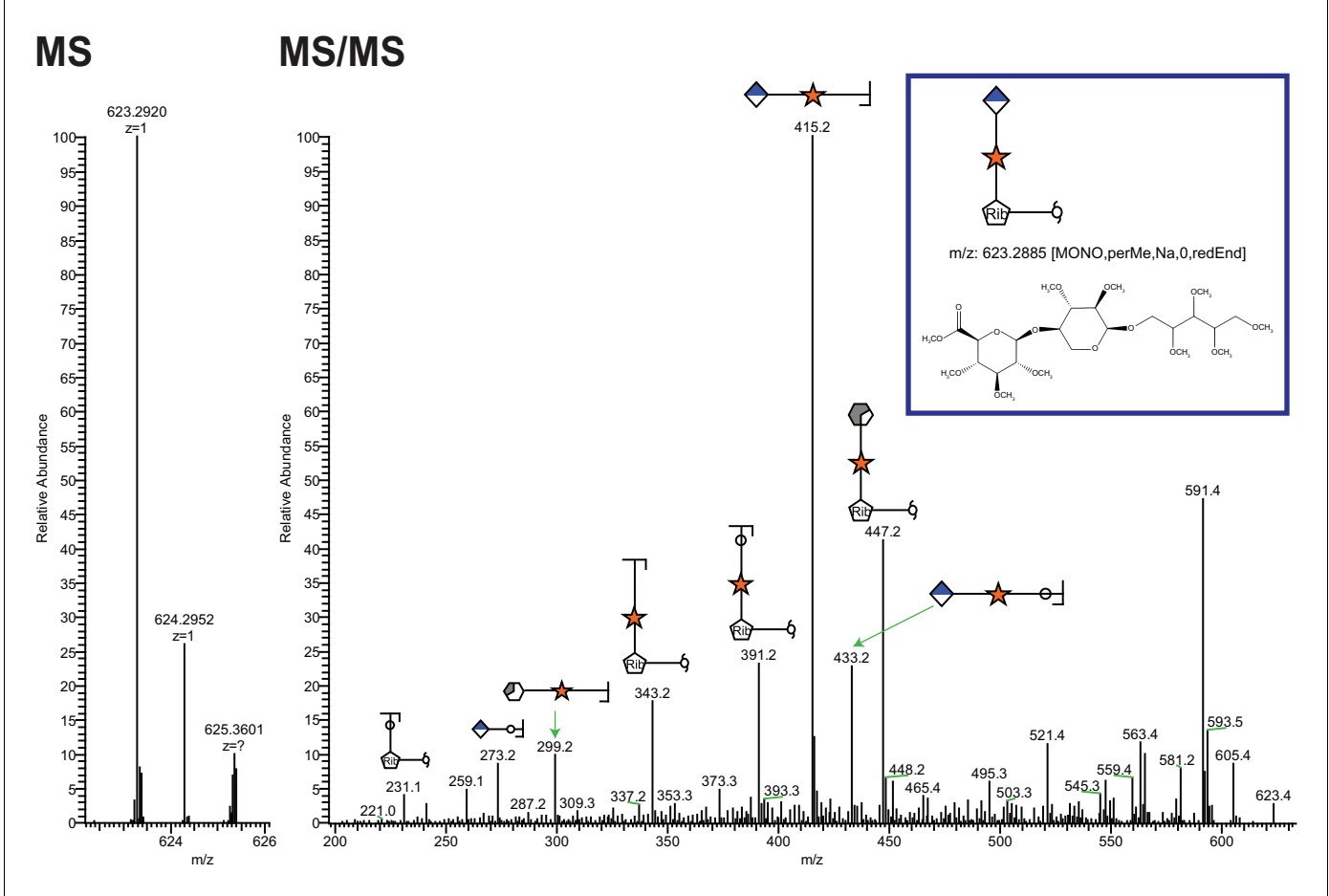

**Figure 2.** Ribitol-Xyl-GlcA is released from α-DG upon cleavage of phosphodiester linkage. Following treatment of purified α-DG-340 with aqueous HF, released glycans were captured, permethylated and analyzed by tandem mass spectrometry. On the left is a zoom view of the full MS scan demonstrating the intact mass and to the right the fragmentation pattern leading to the identification of GlcA-Xyl-pentitol. Experiments were performed in triplicate with identical findings each time.

The following figure supplements are available for figure 2:

**Figure supplement 1.** Extended Ribitol-(Xyl-GlcA)n Structures are also observed.

**Figure supplement 2.** Phosphodiester cleavage but not glycosidase treatment abolishes IIH6 reactivity on α-DG.

purified from whole cell extracts (*Figure 3—figure supplement 1*). The enzyme was capable of generating CDP-ribitol or CDP-ribose using CTP and ribitol-5-phosphate or ribose-5-phosphate, respectively, but was not able to generate the sugar (alcohol) nucleotides with ribitol or ribose (*Figure 3*). Thus, ISPD is a CDP-ribitol (ribose) pyrophosphorylase that generates the needed reduced sugar (alcohol) nucleotide for integration of a ribitol into the functional O-Man glycan structure. During the preparation of our manuscript, a study was published demonstrating that CDP-ribose, CDP-ribitol, and CDP-ribulose could all be generated by mammalian ISPD consistent with our present findings (*Riemersma et al., 2015*).

## Phosphodiester-linked ribitol is connected to the M3 Glycan

Having identified the ribitol directly attached to xylose, we sought to identify whether the ribitol was the bridge between xylose and the phosphotrisaccharide on the polypeptide. We also wanted to confirm the open ring structure enforced by the addition of ribitol, a reduced sugar. In such a linear structure, mild periodate treatment would be expected to cleave vicinal diols and release matriglycan from the polypeptide. Thus, we performed mild periodate cleavage of the purified α-DG-340 protein and demonstrated a loss of reactivity with the IIH6 antibody that recognizes functional LARGE-modified glycoprotein (*Figure 4—figure supplement 2*). Furthermore, following tryptic cleavage, we examined the resulting α-DG peptides by tandem mass spectrometry (*Figure 4—figure supplement 1*). We were able to identify multiple glycopeptides where Thr317 (a known site for M3 glycans) was modified to indicate a cleaved ribitol fragment on the phosphotrisaccharide. We observed pairs of essentially identical glycopeptides that showed cleavage between C2 and C3 or between C3 and C4 (*Figure 4*). We also observed peptides with an additional phosphate suggesting that there can be 2 phosphates on the trisaccharide of which at least one is in a phosphodiester linkage to ribitol (*Figure 4*). Thus, the ribitol appears most likely to be connected at C1 to the oxygen of a phosphate group on the M3 glycopeptide and at the C5 to xylose (note that ribitol C1 and C5 are indistinguishable due to the nature of the reduced sugar and thus nomenclature for closed-ring standard sugar nucleotides is being used to describe linkage).

## TMEM5 is a xylose transferase

In our previous manuscripts describing B4GAT1 (*Praissman et al., 2014*; *Willer et al., 2014*), we had provided evidence for an underlying xylose in the linker region of matriglycan between the phosphotrisaccharide and the LARGE-dependent Xyl-GlcA repeat disaccharide. Among the uncharacterized genes harboring mutations in patients with CMD, TMEM5 shows sequence similarity to glycosyltransferases (*Vuillaumier-Barrot et al., 2012*). *TMEM5* was also among the genes uncovered in a screen for loss of IIH6 binding and Lassa virus entry, readouts that are both dependent on functional glycosylation of α-DG (*Jae et al., 2013*). Thus, we investigated this enzyme as a candidate xylose transferase. We overexpressed and purified the truncated catalytic domain fused to GFP and a His tag (*Figure 5—figure supplement 1A*). Using the UDP-Glo assay, we were able to show that transmembrane deleted dTM-TMEM5 can hydrolyze UDP-Xyl, but not other UDP-sugars (*Figure 5A*) demonstrating selective hydrolytic activity in the absence of acceptor glycans. Furthermore, we were able to show that recombinant full length TMEM5 (*Figure 5—figure supplement 1B*) can be used to label α-DG-Fc340 (Fc-tagged α-DG-340) with radiolabeled UDP-Xyl [Xyl-$^{14}$C] expressed from *TMEM5*-deficient patient cells compared to α-DG-Fc340 mutated at the M3 site (TPT, 317–319, converted to APA) as a negative control (*Figure 5B*). Also, we were able to show that dTM-TMEM5 can use UDP-Xyl to transfer Xyl to CDP-ribitol as the acceptor but not ribitol, ribitol-5-P, or CMP-Neu5Ac suggesting the need for the ribitol to be in a phosphodiester linkage in order to be an acceptor (*Figure 5C*). In summary, these results provide strong, but not direct, evidence that TMEM5 is the xylosyl transferase enzyme for modification of ribitol that is in a phosphodiester linkage to the M3 glycan on α-DG.

## TMEM5 knockdown in zebrafish recapitulates WWS phenotype

The highly conserved *TMEM5* gene, which has formerly been implicated in CMD, encodes a type II transmembrane protein with a predicted glycosyltransferase domain. To investigate the role of TMEM5 in vertebrates, we knocked down the zebrafish (*Danio rerio*) orthologue using antisense morpholino oligonucleotides (MO). The endogenous zebrafish *tmem5* transcript was detected

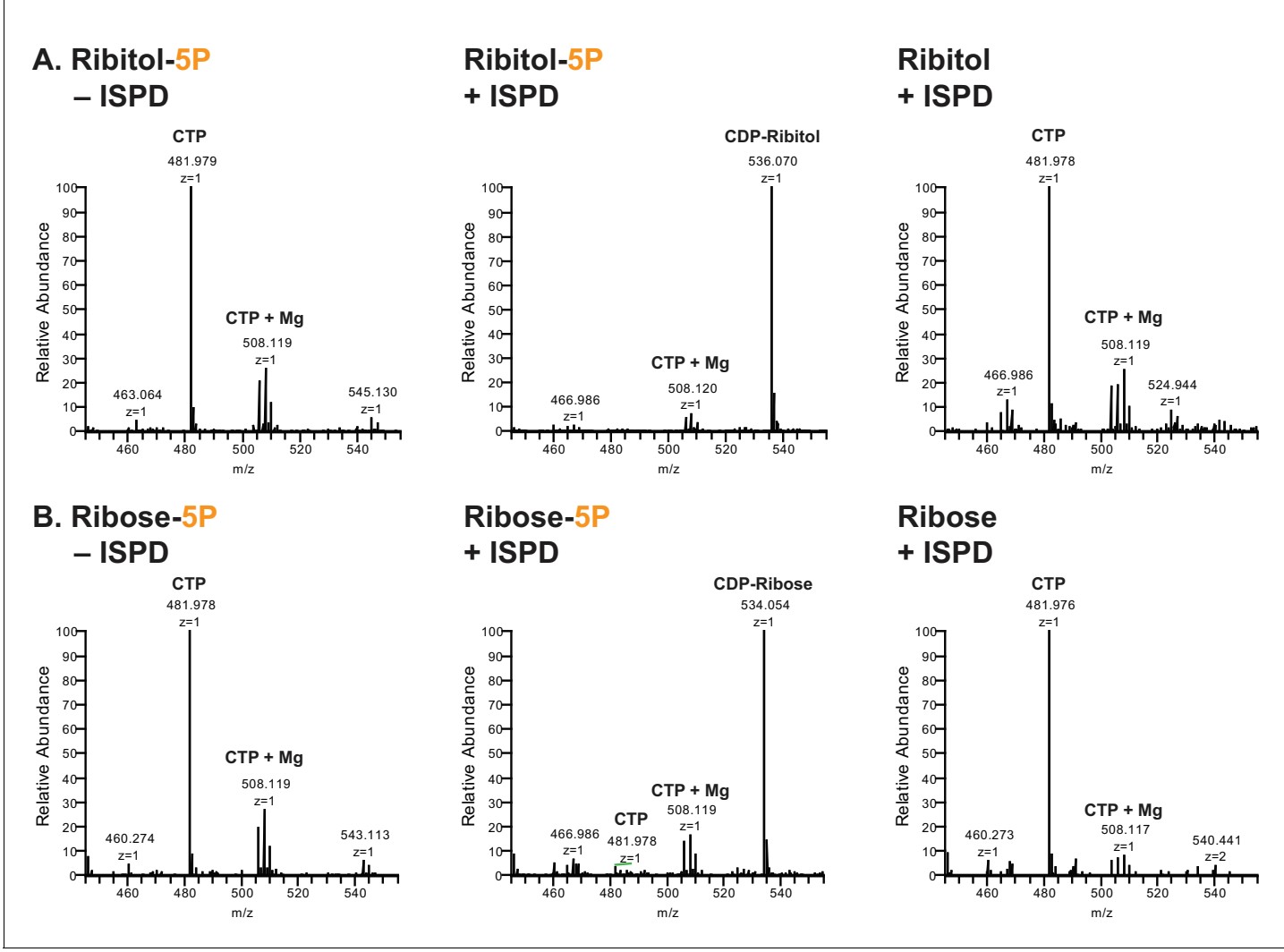

**Figure 3.** ISPD synthesizes CDP-Ribitol from CTP and Ribitol-phosphate. (**A**) Purified recombinant ISPD (+ISPD) or buffer control (-ISPD) was incubated with CTP and either ribitol-5-phosphate (Ribitol-5P) or ribitol (Ribitol) and the donor substrate (CTP, MW 481.98) and/or product (CDP-ribitol, MW 536.07) was detected by mass spectrometry. CDP-ribitol was only observed in the presence of ISPD, CTP, and Ribitol-5P. (**B**) Purified recombinant ISPD (+ISPD) or buffer control (-ISPD) was incubated with CTP and either ribose-5-phosphate (Ribose-5P) or ribose (Ribose) and the donor substrate (CTP, MW 481.98) and/or product (CDP-ribose, MW 534.05) detected by mass spectrometry. CDP-ribose was only observed in the presence of ISPD, CTP, and Ribose-5P. Note that CDP-ribose has a molecular weight that is 2 daltons less than CDP-ribitol. Shown are representative images of an N = 3.

The following figure supplement is available for figure 3:

**Figure supplement 1.** Coomassie stained SDS-PAGE gel of purified ISPD.

throughout early embryonic development (*Figure 6—figure supplement 1*). Injection of *tmem5* MO specifically inhibited the expression of green fluorescent protein-tagged TMEM5 in a dose-dependent manner (*Figure 6—figure supplement 1*). Knockdown of *tmem5* caused an increased percentage of embryos with mild to severe hydrocephalus (95% in total) and significantly reduced eye size, reminiscent of pathological defects in WWS (*Figure 6A*). To test whether the brain and eye abnormalities were caused by MO off-target effects mediated through a p53-dependent cell death pathway (*Robu et al., 2007*), we inhibited p53 activation by co-injection of *p53* MO. 88% of embryos still displayed hydrocephalus and significantly reduced eye size was still observed in embryos co-injected with *p53* and *tmem5* MOs (*Figure 6B*), suggesting that the brain and eye abnormalities were not caused by MO off-target effects. As knockdown of *tmem5* also caused reduced motility and lesions

## (Pyro) QIHAT*PT#PVT#AIGPPT#T#AIQEPPS#R (Na)

**\* = phosphotrisaccharide-ribitol fragment**

**# = potential sites of additional sugars**

| Ribitol Cleavage | Mass | ppm | Additional Hex,HexNAc | Pyro-Glu | Sodiated | Additional Phosphate |
|---|---|---|---|---|---|---|
| 2-3 | 4205.953 | 7.3 | 2 , 3 | N | N | N |
| 2-3 | 4285.910 | 5.2 | 2 , 3 | N | N | Y |
| 2-3 | 4228.953 | -1.0 | 2 , 3 | N | Y | N |
| 3-4 | 4258.963 | 8.7 | 2 , 3 | N | Y | N |
| | | | | | | |
| 2-3 | 4350.968 | 1.7 | 3 , 3 | Y | N | N |
| 2-3 | 4367.992 | 2.3 | 3 , 3 | N | N | N |
| 3-4 | 4397.994 | 1.0 | 3 , 3 | N | N | N |
| 3-4 | 4477.961 | 1.3 | 3 , 3 | N | N | Y |
| | | | | | | |
| 2-3 | 4392.004 | 5.9 | 2 , 4 | Y | N | N |
| 2-3 | 4409.032 | 6.1 | 2 , 4 | N | N | N |
| 2-3 | 4488.9952 | 5.3 | 2 , 4 | N | N | Y |
| 3-4 | 4439.011 | -1.4 | 2 , 4 | N | N | N |

**Figure 4.** Mild periodate cleavage reveals that ribitol is connected to the M3 glycan. Following treatment of purified α-DG-340 under mild periodate conditions, the protein was reduced, alkylated, and digested with trypsin. The resulting peptides were analyzed by tandem mass spectrometry. The α-DG-340 tryptic peptide was detected with multiple glycoforms containing a phosphotrisaccharide glycopeptide. Specifically, at least 12 tryptic peptides were identified at less than 9 ppm mass accuracy that contained the phosphotrisaccharide with a ribitol fragment at threonine 317 (the red T*) in two separate analyses. These peptides differed in additional glycosylation by either hexose (Hex) or HexNAc sugars at additional hydroxyl amino acids (indicated by #). We also observed that the N-terminal glutamine was cyclized to pyroglutamic acid on some peptides (Pyro) and that some glycopeptides were sodiated (Na). Furthermore, we noticed the presence of a second phosphate on some phosphoglycopeptides. The 12 peptides observed are grouped by having the same number of additional Hex and HexNAc glycans (2,3 or 3,3 or 2,4). For all 3 sets of glycopeptides we observed glycopeptides that had ribitol fragments between C2 and C3 as well as C3 and C4 indicating that C2, C3, and C4 are not involved in linkages to other moieties. We also observed for each set of glycopeptides the presence of an additional phosphate.

The following figure supplements are available for figure 4:

**Figure supplement 1.** Tandem MS/MS spectra of glycopeptides.

**Figure supplement 2.** Dot blot analysis reveals loss of IIH6 signal for α-DG treated with mild periodate.

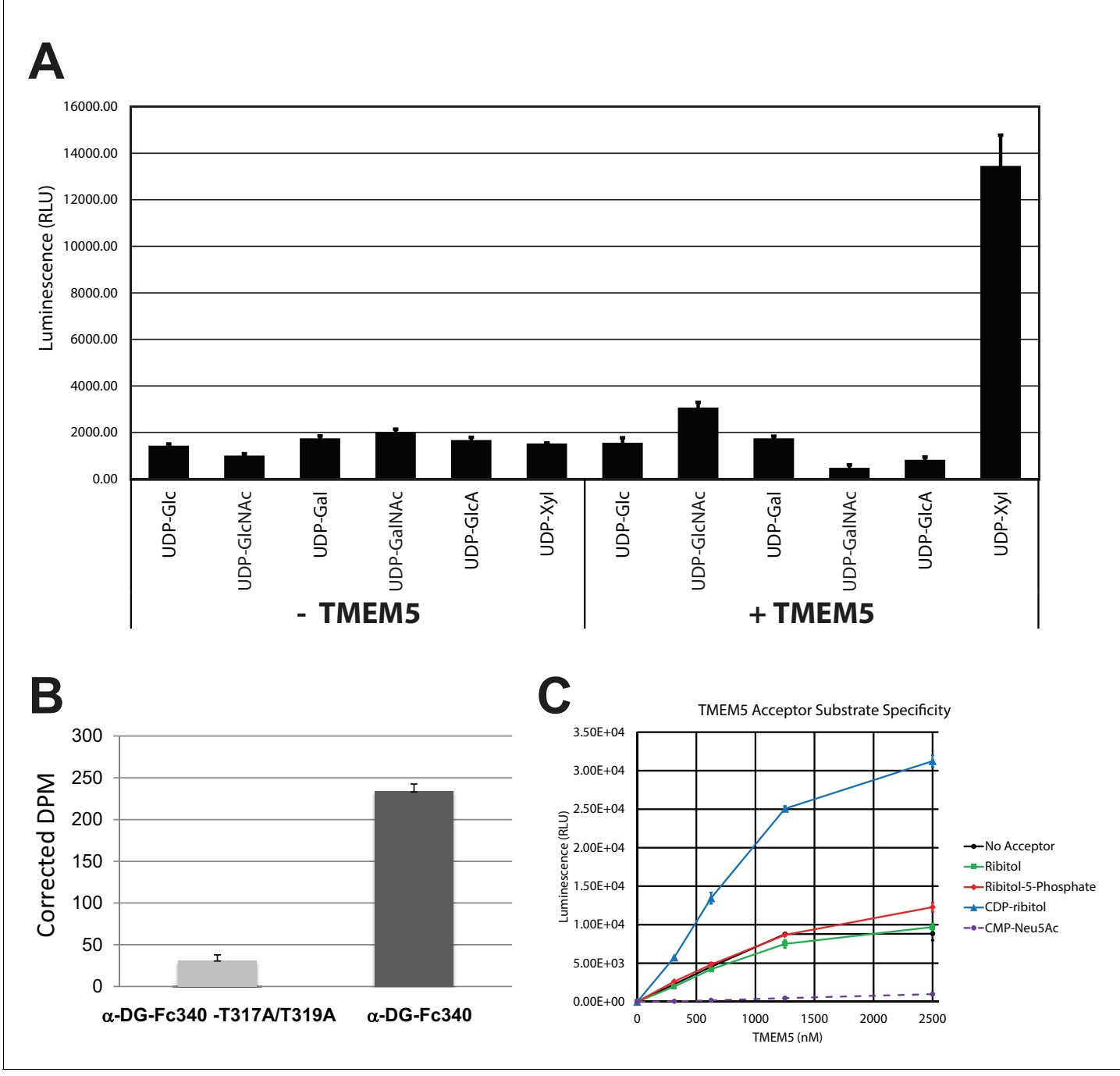

**Figure 5.** TMEM5 is a UDP-Xyl transferase that modifies phosphodiester linked ribitol. (**A**) Various UDP-sugar nucleotides were incubated with (+) or without (-) purified transmembrane deleted and tagged TMEM5 and then analyzed for hydrolysis using the UDP-Glo assay (N = 3). The formation of UDP was only observed when dTM-TMEM5 was incubated with UDP-Xyl. (**B**) Recombinant purified full length TMEM5 and radiolabeled UDP-Xyl [Xyl-$^{14}$C] was incubated with either purified α-DG-Fc340 or mutant α-DGFc340 where the sites for M3 addition had been mutated to alanines (α-DG-Fc340-T317A/T319A) and incorporation of radioactivity into the proteins were determined by scintillation chromatography (N = 3). α-DGFc340 that contains the sites for M3 addition is the preferred substrate for addition of radiolabeled Xyl by TMEM5. (**C**) dTM-TMEM5 (2500, 1250, 625, or 312.5 nM) and 50 μM UDP-Xyl donor substrate were incubated without or with 1 mM of the indicated acceptor molecules in 0.1 M MES pH 6.0 and 10 mM MgCl$_2$ for 18 hr at 37°C, and the release of UDP was detected by the UDP-Glo assay. RLU: Relative Luminescence Units. Data are represented as the mean of three trials and error bars represent standard deviation.

The following figure supplements are available for figure 5:

*Figure 5 continued on next page*

*Figure 5 continued*

**Figure supplement 1.** Analysis of purified TMEM5 constructs.

**Figure supplement 2.** Specificity of TMEM5 for CDP-ribitol as an acceptor.

in the myotome (data not shown), we assessed the sarcolemma integrity using Evans Blue dye (EBD), which does not penetrate into intact muscle fibers. Muscle fibers were infiltrated by EBD before undergoing degeneration (*Figure 6C*), suggesting a pathological mechanism in which knockdown of *tmem5* leads to compromised sarcolemma integrity. As defective glycosylation of α-DG is a pathological hallmark of WWS, we tested whether knockdown of zebrafish *tmem5* would affect the glycosylation of α-DG. Compared to control embryos, knockdown of *tmem5* caused a 44% reduction of glycosylated α-dystroglycan (IIH6 epitope) on Western blots (*Figure 6D*). Together, these results clearly illustrate a role for TMEM5 in functional glycosylation of α-DG and knockdown of this enzyme generating a CMD phenotype in vertebrates.

## Identification of a new *TMEM5* missense mutation in a family with WWS

Previously, it was reported that mutations in *TMEM5* can cause WWS, a congenital form of muscular dystrophy with severe brain involvement (*Vuillaumier-Barrot et al., 2012*; *Jae et al., 2013*). To investigate if one of the unidentified consanguineous WWS families was affected by a mutation in *TMEM5* we performed linkage analysis and whole exome sequencing (WES) in three siblings (*Figure 7—figure supplement 1A*). Genomic DNAs from the three siblings were genotyped and the call rates for the genotyping were 99.7%, 96.0% and 91.9% for 02243-d (P1), 02243-a (p2) and 02243-b, respectively. Using ~69K SNPs that overlapped between the two platforms, homozygosity-by-descent (HBD) analysis was performed. All three samples had multiple long (>10 cM) stretches of homozygous genotypes, confirming that they were descendants of a consanguineous marriage. Four regions that were homozygous in the two affected siblings, but heterozygous or homozygous for the other alleles in the unaffected sibling, were identified (*Figure 7—figure supplement 1B*). All 6,647 coding exons from these four intervals were subject to targeted sequencing in all three samples.

The genomic library was sequenced and variant filter strategies were applied and retained for variants on chromosome 12 and chromosome X. A homozygous, non-reference c.997G>A (p.G333R) sequence variant was found in the *TMEM5* gene (Chr. 12) of the two affected siblings (P1 and P2), while the unaffected sibling was homozygous for the wild type sequence (c.997G, p.G333) (*Figure 7—figure supplement 1A*). The variant was predicted to be damaging by PolyPhen-2 (*Adzhubei et al., 2013*) and SIFT (*Sim et al., 2012*). Sanger sequencing confirmed this new deleterious variant in exon 6 of *TMEM5* (*Figure 7—figure supplement 1C*). The highly conserved affected amino acid p.G333 is located in a functional domain that is predicted to have glycosyltransferase activity (*Figure 7—figure supplement 1D,E*).

The new p.G333R *TMEM5* mutation was identified in a consanguineous family of Pakistani descent. Two out of three pregnancies resulted in fetuses with WWS (P1 and P2), while the third child is normal (*Figure 7—figure supplement 1A*). Both WWS fetuses had classical brain developmental abnormalities detected by prenatal ultrasound (see *Figure 7—figure supplement 2A*). The first pregnancy (P1) went to term with the child passing away at 4 months of age. The second pregnancy was terminated at 18 weeks (P2); a quadriceps skeletal muscle sample utilized in our studies was obtained after termination. Severe dystrophic features were noted in cryosections of the muscle, and immunofluorescence studies showed an abnormal pattern of dystrophin-glycoprotein complex expression characteristic of a severe dystroglycanopathy (*Figure 7—figure supplement 2B*).

Although patients with *TMEM5* mutations have been reported before (*Vuillaumier-Barrot et al., 2012*; *Jae et al., 2013*) the α-DG glycosylation status in these patients has not been investigated. Immunofluorescence and western blot analysis of skeletal muscle from the 18 week fetus (P2) showed an α-DG glycosylation defect similar to previously described glycosylation-deficient WWS patients (*Willer et al., 2012*) with complete loss of both functional glycosylation and laminin binding (*Figure 7—figure supplement 2B,C*). The loss of receptor function and shift of α-DG core protein to lower molecular weight was confirmed in skeletal muscle (P1, *Figure 7—figure supplement 2C*), as

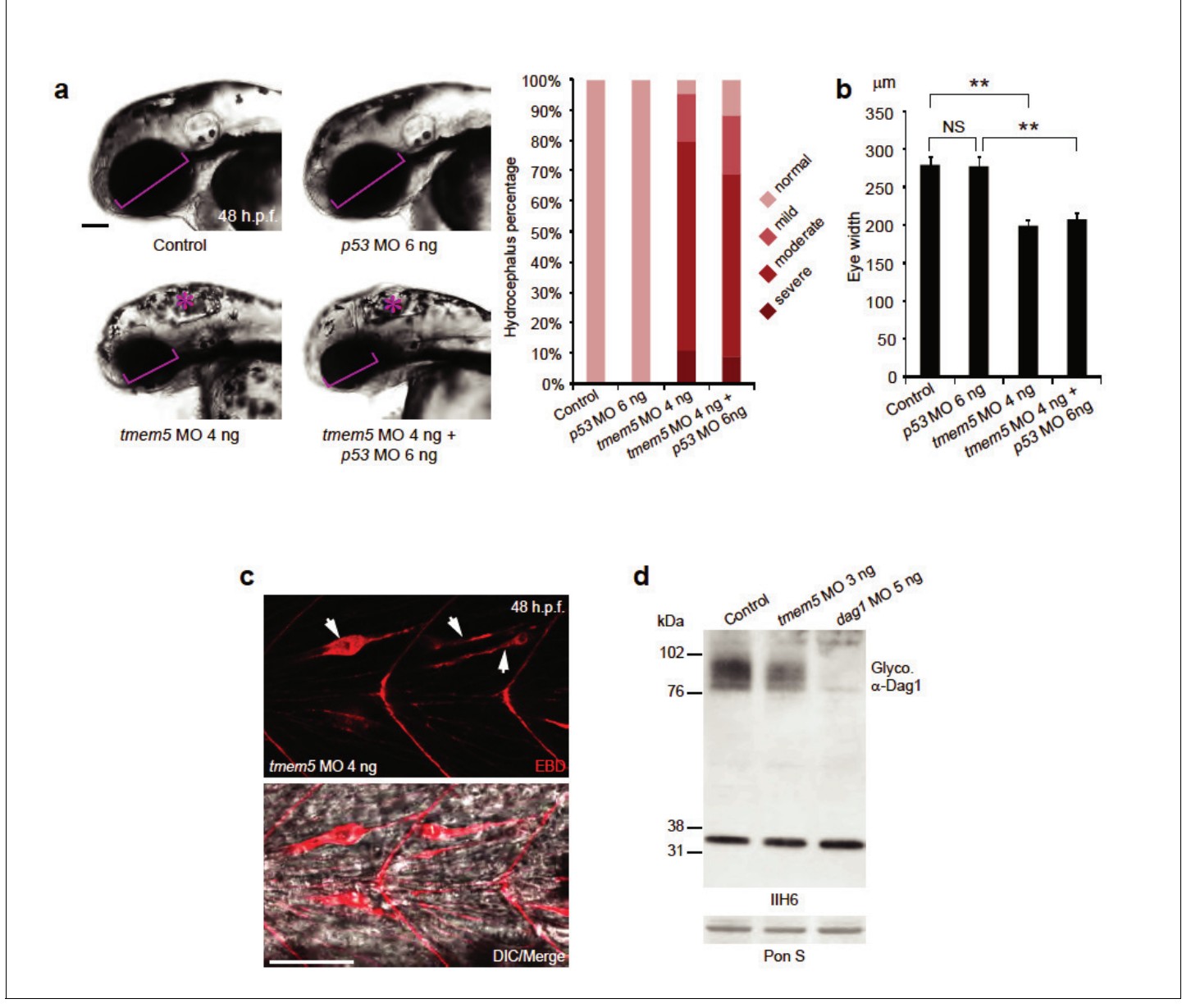

**Figure 6.** Knockdown of zebrafish *tmem5* recapitulates characteristic defects in WWS. (A) Compared with uninjected control and *p53* MO injected embryos, embryos injected with *tmem5* MO alone or together with *p53* MO developed mild to severe hydrocephalus (asterisk) from 32 hr post fertilization (h.p.f.) (95% and 88%, respectively). Each bar represents a combination of two independent blindly-scored experiments. n = 114–150 embryos. Scale bar, 100 μm. (B) One-way ANOVA analysis revealed that *tmem5* MO injected embryos have significantly reduced eye size (bracket in panel a) at 48 -h.p.f. Inhibition of MO off-target effects by co-injection of *p53* MO does not rescue the reduced eye size. n = 15 embryos in each group. Error bars, s.d. **p< 0.01. NS, not significant. (C) Knockdown of *tmem5* compromised the sarcolemma integrity as indicated by EBD infiltrated muscle fibers (arrows) at 48 h.p.f. Muscle fibers undergoing degeneration pull away from chevron-shaped myosepta. Scale bar, 50 μm. (D) Compared with uninjected control, western blotting with IIH6 antibody showed a 44% reduction in glycosylated α-dystroglycan in *tmem5* MO-injected embryos at 48 h. p.f. Equal loading was demonstrated by Ponceau S and unknown glycoproteins (~35 kDa) detected by IIH6 in all lanes.

The following figure supplement is available for figure 6:

**Figure supplement 1.** Zebrafish *tmem5* temporal gene expression and MO specificity.

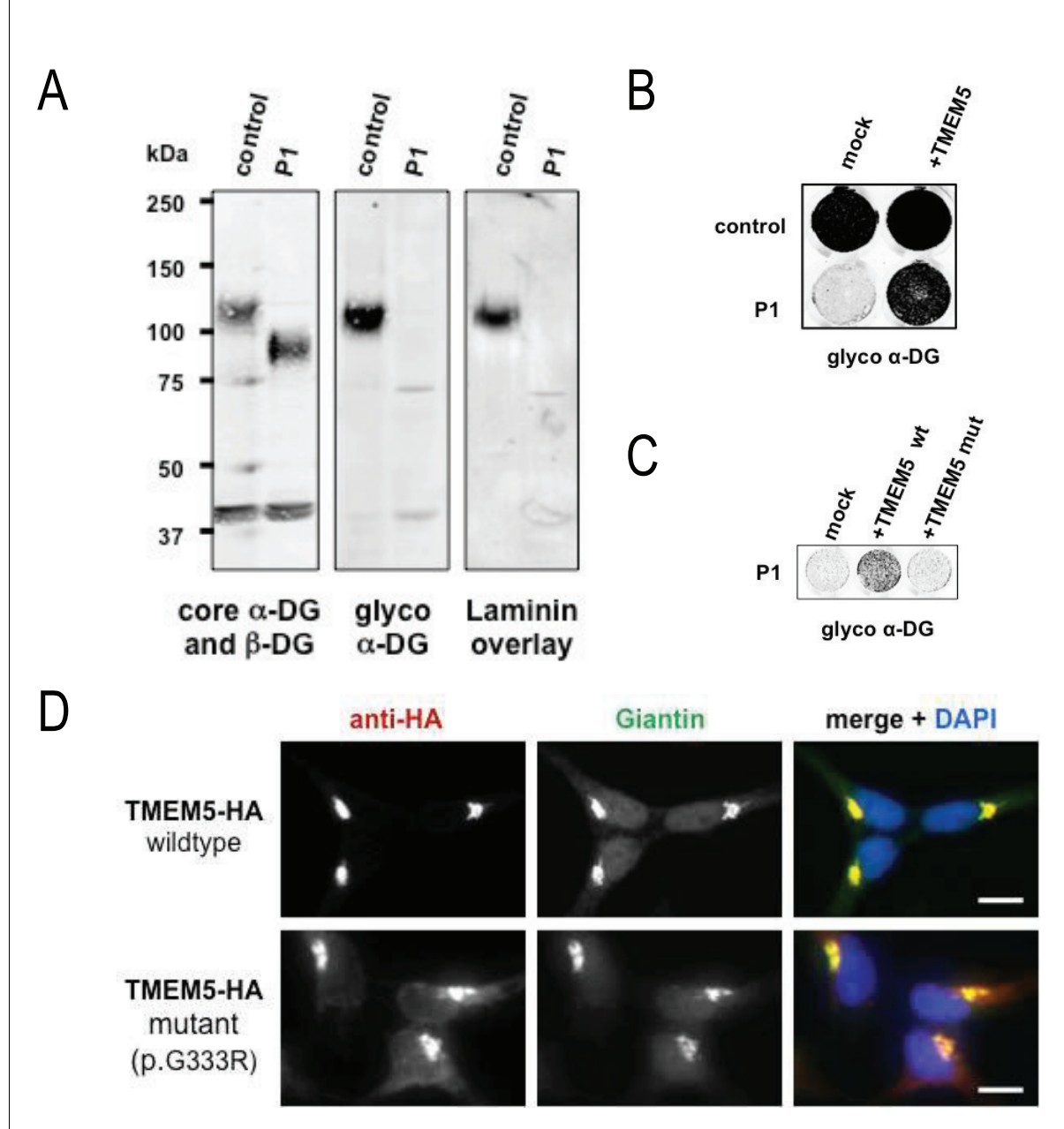

**Figure 7.** A missense mutation of *TMEM5* in a WWS family leads to a loss of functional α-DG. (A) Analysis of α-DG glycosylation status in *TMEM5*-deficient WWS patient (P1) dermal fibroblasts. WGA-enriched cell lysates from control and *TMEM5* P1 fibroblasts were subjected to biochemical analysis. α-DG glycosylation status as assessed by western blotting using antibodies against the glycosylated form of α-DG (IIH6), core α-DG (G6317), and by laminin overlay assay (N = 3). An antibody against β-DG (AP83) was used to assess loading. Apparent molecular masses are indicated. (B) On-Cell western-based complementation assay (N = 3) of *TMEM5*-WWS patient (P1) fibroblasts after adenoviral gene transfer with wildtype TMEM5 expression construct. Rescue of α-DG functional glycosylation was monitored with α-DG glyco (IIH6) antibodies. (C) On-Cell western-based complementation assay (N = 3) of P1 fibroblasts after nucleofection with a wild-type or mutant (p.G333R) human TMEM5 expression construct. Rescue of α-dystroglycan functional glycosylation was detected with antibody to glycosylated α-dystroglycan (glyco α-DG, IIH6). (D) Subcellular localization of HA-tagged wildtype and WWS mutant TMEM5 protein, as assessed by immunofluorescence. HEK293T cells stably expressing HA-tagged proteins were stained with anti-HA (red), anti-Giantin (Golgi marker, green) and 4´,6-diamidino-2-phenylindole (DAPI, nuclei, blue). Individual stainings for HA and Giantin are shown in greyscale and a merged image is shown in color. Scale bars indicate 10 µm.

The following figure supplements are available for figure 7:

**Figure supplement 1.** Identification and validation of *TMEM5* as the disease gene in a WWS family.

*Figure 7 continued on next page*

*Figure 7 continued*
**Figure supplement 2.** Clinical presentation and α-DG glycosylation defect in *TMEM5*-WWS patient P2.

well as in cultured skin fibroblast samples (P2, *Figure 7A*). To demonstrate the pathogenicity of the identified *TMEM5* mutations, we conducted complementation assays on skin fibroblasts derived from the first child (P1). In the patient cells, expression of wildtype *TMEM5* fully restored functional glycosylation while the p.G333R mutant protein did not (*Figure 7B*). Functional rescue of patient cells supports the interpretation that TMEM5 p.G333R has pathogenic relevance and causes WWS. Furthermore, we determined whether the identified TMEM5 p.G333R variant affects expression and localization of the mutant protein. HA-tagged wildtype and p.G333R *TMEM5* constructs were transfected into in HEK293 cells. Immunofluorescence and co-localization with a Golgi marker Giantin confirmed that both proteins are expressed and localize to the Golgi apparatus without significant ER mislocaliztion (*Figure 7D*). This result shows that the loss of TMEM5 function in the WWS patients

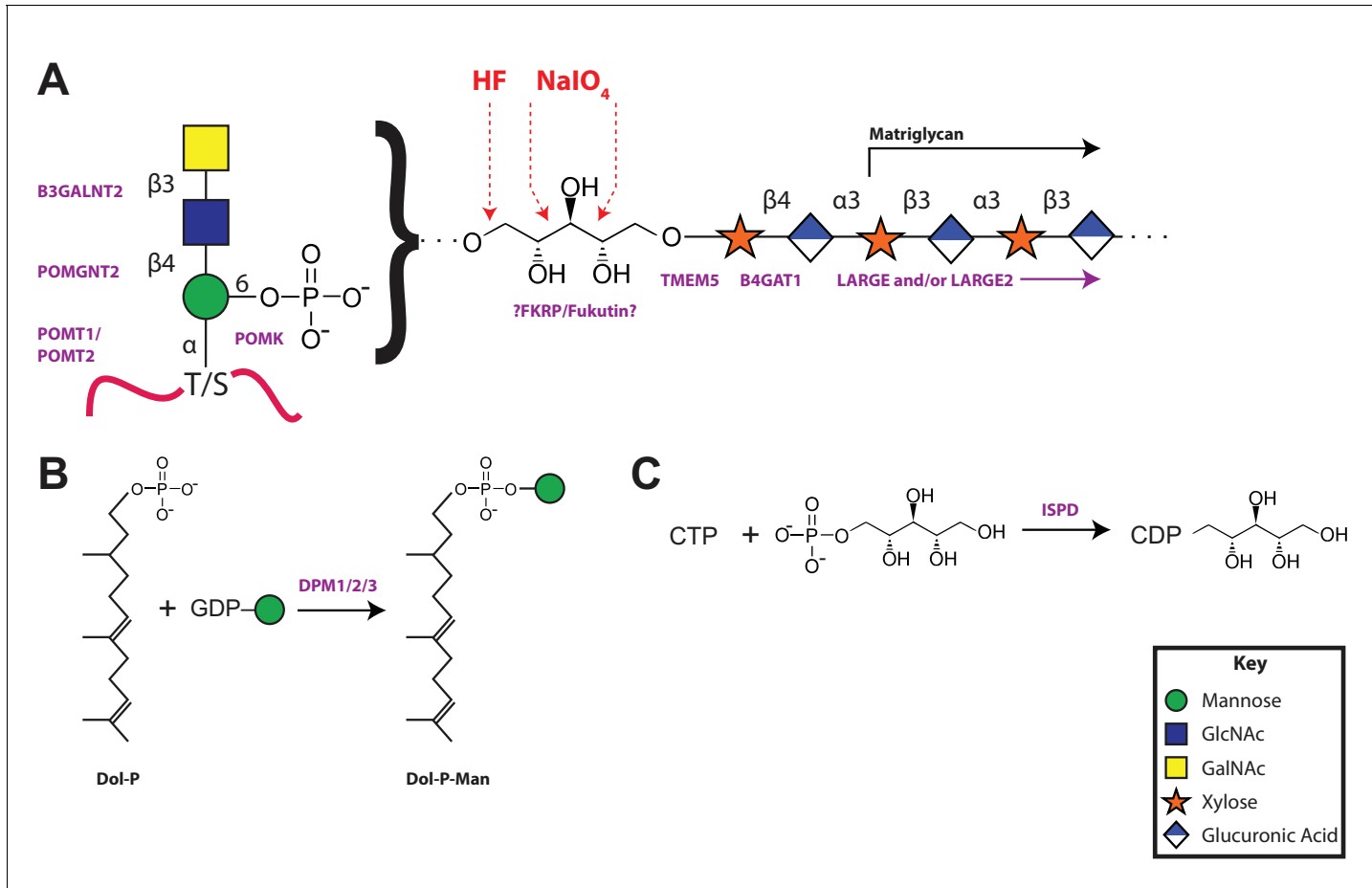

**Figure 8.** A model of the α-DG functional glycan structure and enzymes that contribute to its synthesis. (**A**) The proposed functional M3 glycan structure is shown with known or putative (?FKRP/Fukutin?) enzymes involved labeled in purple that when defective lead to secondary dystroglycanopathies. Also detailed in red are the observed sites of ribitol cleavage by aqueous HF (HF) and mild periodate (NaIO₄) treatment. (**B**) The DPM1/2/3 enzyme complex (labeled in purple) is required for the synthesis of Dolichol-phosphomannose (Dol-P-Man) from GDP-Man and Dolichol phosphate (Dol-P). (**C**) ISPD (labeled in purple) is required for the synthesis of CDP-ribitol from CTP and ribitol-phosphate. Defects in DPM1/2/3 and ISPD are considered tertiary dystroglycanopathies since they synthesize the sugar donors Dol-P-Man (POMT1/2) and CDP-ribitol (presumably Fukutin and/or FKRP) for subsequent glycosyltransferases.

is not caused by a cellular processing defect, but rather directly affects the catalytic domain and abrogates the proposed xylose glycosyltransferase activity.

## Model of the proposed α-DG functional glycan structure and enzymes that contribute to its synthesis

Collectively our data supports a model of a functional glycan structure on α-DG where the phosphorylated trisaccharide, which we previously identified (*Yoshida-Moriguchi et al., 2010*), is extended by a ribitol (by an unknown enzyme(s) but presumably Fukutin and/or FKRP) in a phosphodiester linkage followed by the addition of a priming Xyl (added by TMEM5) and GlcA (added by B4GAT1) before extension with the repeating disaccharide matriglycan by LARGE (*Figure 8A*). Mutations in all of the enzymes in this pathway have been demonstrated to generate secondary dystroglycanopathies in patients (*Wells, 2013*; *Yoshida-Moriguchi and Campbell, 2015*; *Dobson et al., 2013*; *Endo, 2015*; *Stalnaker et al., 2011*). Further, ISPD, similar to the DPM1/2/3 enzyme complex that generates dolichol-phosphomannose (*Figure 8B*) for initial mannosylation, generates CDP-ribitol (*Figure 8C*) for ribosylation of the phosphotrisaccharide. Both of these processes are involved in the generation of donors for the enzymes involved in functional glycosylation of α-DG and as such CMD resulting from these enzymes should then be referred to as tertiary dystroglycanopathies (*Vuillaumier-Barrot et al., 2012*; *Lefeber et al., 2009*).

## Discussion

Initiation and further extension of *O*-Man glycans on α-DG is required for proper recognition by ECM proteins (*Yoshida-Moriguchi and Campbell, 2015*; *Dobson et al., 2013*; *Endo, 2015*; *Endo and Manya, 2006*). Failure of proper glycan elaboration is causal for a significant subset of congenital muscular dystrophies (CMD) referred to as dystroglycanopathies that range from severe Walker-Warburg syndrome (WWS) to the much milder Limb-Girdle muscular dystrophy (LGMD) presumably resulting from the severity of the mutation on enzyme expression, stability, localization, and/or activity (*Wells, 2013*; *Stalnaker et al., 2011*; *Muntoni et al., 2011*). We and others have worked for the last two decades on elucidating the functional *O*-Man glycan structures that are essential for effective interactions with extracellular matrix proteins and defective in CMD (reviewed recently in *Praissman and Wells, 2014*). Here, we have further elucidated the functional glycan structure and attempted to assign the enzymes responsible for each step in the biosynthetic pathway.

This work extended our previous studies demonstrating a phosphodiester linkage bridging from the phosphotrisaccharide to the extended LARGE-dependent Xyl-GlcA repeat on α-DG (*Yoshida-Moriguchi et al., 2010*). We previously identified the M3 phosphotrisaccharide glycan structure following aqueous HF treatment and assigned its sites of modification on the α-DG polypeptide (*Yoshida-Moriguchi et al., 2010*). In the present study we examined the structure of the glycan released from polypeptide upon HF cleavage of the phosphodiester linkage (*Figure 8*). To simplify the analysis, the length of the LARGE-dependent repeat was reduced by overexpression of a small IIH6-reactive fragment of α-DG (28 amino acids) containing only one M3 site (TPT, 317–319) as a fusion protein in HEK293F cells, which express low levels of LARGE. Following HF release, we were able to identify pentitol-Xyl-GlcA by tandem mass spectrometry, as well as further extended versions (*Figure 2*).

We were somewhat surprised to find pentitol (a reduced sugar polyol) attached to the xylose. Assuming that this reduced sugar was likely transferred as an activated sugar nucleotide, we attempted to identify a mammalian CDP-ribitol pyrophosphorylase. ISPD was a reasonable candidate since, as a putative cytosolic nucleotidyltransferase (*Vuillaumier-Barrot et al., 2012*), it is mutated in a subset of CMD and LGMD patients (*Willer et al., 2012*; *Cirak et al., 2013*), and bacterial homologs participate in the synthesis of CDP activated alcohols (*Baur et al., 2009*; *Richard et al., 2001*). Purified recombinant enzyme was able to synthesize CDP-ribitol from ribitol-phosphate and CTP demonstrating that it is indeed a CDP-ribitol pyrophosphorylase (*Figure 3*). During the preparation of our manuscript, a study was published demonstrating that CDP-ribose, CDP-ribitol, and CDP-ribulose could all be generated by mammalian ISPD consistent with our present results (*Riemersma et al., 2015*). These findings also lend themselves to considering the source of ribitol-P in mammals that likely is generated from ribose-5-phosphate, from the pentose phosphate

pathway, by the action of aldose reductase (*Perl et al., 2011*). Alternatively, given that ISPD can act on ribitol-phosphate and ribose-phosphate, it is possible that the reduction occurs after the formation of the sugar nucleotide CDP-ribose to form CDP-ribitol. Further investigation of kinetic constants and cellular abundances need to be pursued to determine the order of events. Furthermore, no CDP-ribitol transporter into the secretory pathway has been identified yet. Given the facts that mutations in SLC35A1, a proposed CMP-sialic acid transporter, have been implicated in decreased functional glycosylation of α-DG that is independent from sialic acid (*Jae et al., 2013*; *Riemersma et al., 2015*) and that sugar nucleotide transporters often have higher selectivity for the nucleotide than the sugar, it is inviting to speculate that SLC35A1 is a CDP-ribitol transporter though this remains to be formally tested.

Given that ISPD could generate CDP-ribose and CDP-ribitol we wanted to confirm that it was ribitol and not ribose that was transferred into the functional *O*-Man glycan on α-DG. Further, we wanted complementary experimental evidence that would be consistent with the phosphodiester cleavage study that the ribitol was connected directly to the phosphotrisaccharide of α-DG. Both reduced and non-reduced glycans are susceptible to cleavage by mild periodate treatment between vicinal diols (*Collins et al., 1997*). If ribitol was indeed inserted between the phosphotrisaccharide and the priming xylose, one would expect loss of IIH6 reactivity from the α-DG-340 fusion protein as we observed (*Figure 4—figure supplement 2*). Furthermore, mass spectrometry analysis of the resulting peptides revealed cleavage between carbons 2–3 and between 3–4 of the ribitol (*Figure 4*). We also observed phosphotrisaccharide glycopeptides with an additional phosphate moiety suggesting that the M3 trisaccharide can have 2 phosphates connected directly to it (one is at the 6-position of Man and the other we could not resolve). These data are highly suggestive that an M3 phosphate oxygen is connected to the C1 carbon while the xylose is connected to the C5 carbon of the ribitol (note that given the structure of ribitol, C1 and C5 are equivalent and thus we have chosen to use a numbering system that is most consistent with other sugar nucleotides) (*Figure 8*).

Based on our proposed structure, we also predicted a xylose transferase to further elongate the structure and serve as a substrate for B4GAT1 extension with GlcA as a primer for LARGE addition of matriglycan. Given that mutations in *TMEM5* have recently been described as being causal for CMD and that the putative enzyme shows sequence homology to glycosyltransferases (*Vuillaumier-Barrot et al., 2012*; *Jae et al., 2013*), we explored the possibility that TMEM5 was a xylose transferase (*Figure 5*). We were able to confirm that recombinant TMEM5 was able to hydrolyze UDP-Xyl but not other UDP-sugars. We also observed that while ribitol and ribitol-5-P were not acceptors in vitro that CDP-ribitol was an acceptor for TMEM5 catalyzed transfer of Xyl from UDP-Xyl likely due to mimicking the phosphodiester linkage of the ribitol in the glycopeptide structure. Further, recombinant TMEM5 was able to transfer radiolabeled Xyl from UDP-Xyl to α-DG-Fc340 expressed in a CMD patient cell line with a homozygous mutation in *TMEM5* but was not able to effectively transfer to α-DG-340 when the putative M3 sites (TPT, 317–319, converted to APA) were eliminated. This data taken together strongly suggests that TMEM5 adds Xyl to ribitol when the ribitol is in a phosphodiester linkage. Given the scarcity of studies on TMEM5, we further characterized the functional relevance of TMEM5 in a zebrafish knockdown model and observed a severe CMD phenotype (*Figure 6*). Further, we identified a novel p.G333R TMEM5 mutation in a consanguineous family with 2 children affected by WWS. We demonstrated that wildtype TMEM5 but not the p.G333R TMEM5 mutant construct could complement the mutation in the patient cell line with regards to functional α-DG (*Figure 7B,C*). Subcellular localization studies showed normal expression and localization of the mutant protein, suggesting that the p.G333R missense mutation in the predicted catalytic site likely interferes with enzymatic activity (*Figure 7D*). Thus, we propose that TMEM5 modifies the ribitol in a phosodiester linkage with a xylose and B4GAT1 then extends the structure with a GlcA to serve as a primer for LARGE extension with the repeating Xyl-GlcA disaccharide, matriglycan.

In summary, it appears that synthesis of the functional glycan on α-DG requires nearly a dozen enzymes, many of which presumably are exclusively used on a select set of sites on α-DG (*Figure 8*). While we have further elucidated a functional glycan structure required for interaction with extracellular matrix proteins, several questions remain. For instance, while we have postulated that FKRP and/or Fukutin are involved in the transfer of ribitol using CDP-ribitol as the donor, we have not formally tested this activity due primarily to technical difficulties with expressing these two proteins. Also, while we have confirmed that TMEM5 is a xylose transferase that can add to a ribitol that is in a phosphodiester linkage, we have not experimentally validated its transfer to a ribitol-

phosphotrisaccharide modified peptide for the same reasons as above. Furthermore, the generation of the functional glycan in quantities necessary to assign anomeric configurations and confirm linkages in the structure is still required. During the revision of this manuscript, a manuscript was published that addresses many of our outstanding questions (*Kanagawa et al., 2016*). In particular, Toda and colleagues were able to establish that Fukutin acts to add a ribitol-P in a phosphodiester linkage to the GalNAc of the phosphotrisaccharide that is then extended with an additional ribitol-P by FKRP before elaboration with Xyl by an unknown enzyme (shown here to be TMEM5) (*Kanagawa et al., 2016*). All of our data presented here is completely consistent with the structure proposed by Toda and colleagues.

In future studies, we would like to determine if there are other sites of functional glycosylation beyond the 317/319 and 379/381 sites on α-DG that we have previously identified (*Yoshida-Moriguchi et al., 2013*; *Yoshida-Moriguchi et al., 2010*). It is also puzzling that while many sites on α-DG and even on other proteins, such as cadherins (*Vester-Christensen et al., 2013*), are *O*-Man modified, only a select few sites on α-DG appear to become modified with M3 glycans containing matriglycan for interaction with ECM proteins. The evolution of such a complex biosynthetic pathway for a few sites on a single protein is an enigma. Understanding this exclusivity as well as the role of M1 and M2 glycans in biology is a major future challenge. In closing, we have defined the role of additional enzymes in the *O*-Mannosylation pathway and further elucidated a functional glycan structure on α-DG for binding to ECM proteins that is lacking in many secondary and tertiary dystroglycanopathies.

## Materials and methods

### Materials

Chemical reagents were primarily purchased from Sigma-Aldrich (St. Louis, MO) at reagent grade or better. Microcon centrifugal filters were purchased from EMD Millipore (Billerica, MA) as were IIH6C4 antibody stocks. Liquid chromatography and mass spectrometry systems were purchased from Thermo Scientific (Waltham, MA). Glycosidases were purchased from ProZyme (Hayward, CA). Spin columns were purchased from Nest group (Southborough, MA). Software was from Thermo Scientific, Protein Metrics (San Carlos, CA) and EuroCarbDB (*Damerell et al., 2015*).

### α-DG-340 preparation and purification

A truncated form of recombinant α-DG (residues 1–340 of human DAG1, Uniprot Q14118) was expressed by transient transfection of suspension culture HEK293F cells as a soluble secreted fusion protein (lacking amino acids 1–312 following furin cleavage in the secretory pathway). The fusion protein coding region was designed, codon optimized, and chemically synthesized by Life Technologies (Thermo Fisher Scientific, Waltham, MA) and subcloned into a mammalian expression vector (*Barb et al., 2012*) that provided a COOH-terminal fusion of the 7 amino acid recognition sequence of the tobacco etch virus (TEV) protease (*Tözsér et al., 2005*), the 'superfolder' GFP coding region (*Pédelacq et al., 2006*), an AviTag recognition site for in vitro biotinylation (*Beckett et al., 1999*), followed by an 8xHis tag (construct designated α-DG340-pGEc2). The vector employs a CMV-based promoter and enhancer sequences to drive recombinant protein expression and the $NH_2$-terminal signal sequence of α-dystroglycan to target entry into the secretory pathway and secretion from the cell. Recombinant expression of the α-DG340 fusion protein in HEK293F cells (Freestyle 293-F cells, ThermoFisher Scientific, verified by RNA-Seq and tested routinely for mycoplasma contamination by PCR, not on the commonly misidentified cell line registry) and purification of the protein from the conditioned medium by $Ni^{2+}$-NTA chromatography was carried out as previously described (*Meng et al., 2013*). Briefly, cells were pelleted by centrifugation at 1000 x g for 10 min, 5 days post-transfection, and the cell culture medium containing GFP-α-DG340- was subjected to Ni-NTA chromatography. GFP-hTMEM5 was eluted with 300 mM imidazole. Fractions containing GFP-α-DG340 were pooled, buffer exchanged into phosphate-buffered saline (PBS, pH 7.2) and concentrated to 1 mg/mL using an Amicon Ultra-15 Centrifugal Filter Unit equipped with a 30,000 NMWL membrane (EMD Millipore).

## α-DG post-phospho moiety release and purification

Purified α-DG-340 (~100 μg) that only contains AA 313–340 of α-DG was buffer exchanged into Milli-Q water using an EMD Millipore 10 kDa NMWL Microcon centrifugal filter according to manufacturer's instructions (20 min centrifugation, 13,500 x g, 25°C, final ratio 1:250). Final imidazole concentration and protein concentration were assessed by NanoDrop ND-1000 spectroscopy (A230 and A280). The purified protein preparation was dried down in a SpeedVac and treated with cold aqueous 48% hydrofluoric acid (Sigma) on ice at 4°C overnight to cleave phosphodiester bonds. HF was removed by drying with $N_2$ gas on ice. Residual trace HF was removed by resuspending with 100 μl Milli-Q $H_2O$ several times and SpeedVac drying (*Yoshida-Moriguchi et al., 2010*). Released glycans were separated from protein by C18 reverse-phase desalting using Nest Group Macrospin columns with 0.1% formic acid (flow-through and washes containing released glycans).

## Glycan permethylation

Permethylation was carried out as described (*Kumagai et al., 2013*). The dichloromethane and aqueous fractions were each analyzed separately.

## Full MS and total ion monitoring of permethylated glycans

Permethylated glycans were resuspended in ~30 μl of 50% MeOH with 1 mM NaOH and loaded into a Hamilton syringe for direct infusion at 0.5 μl/min into an Orbitrap XL (Thermo Scientific). Orbitrap full MS scans and total ion monitoring (TIM) data were acquired for the organic fraction and separately for the aqueous fraction from the permethylation procedure. Full MS scans were acquired for 30 s in m/z range 300–2000 and separately in m/z range 600–2000 with AGC target 2e5, spray voltage 2 kV in positive mode. The TIM method consisted of ion trap scans in positive mode from 400–2000 with parent mass step 2.0, CID activation, isolation width 2.2, 38 normalized collision energy and ITMS MSn AGC target 1e4. Data were analyzed and annotated using a combination of Chem-Draw Professional 15.0 with additional structure information from pubchem.ncbi.nlm.nih.gov and Glycoworkbench 2.1 (*Damerell et al., 2015*).

## Mild periodate treatment

Mild periodate treatment was carried out according to protocols published previously (*Collins et al., 1997*). Purified α-DG-340 (~200 μg) was buffer exchanged into PBS pH 7.2 using an EMD Millipore 10 kDa NMWL Microcon centrifugal filter as above. An aliquot was removed for mock treatment (omission of periodate only). NaIO4 in Milli-Q $H_2O$ was added to achieve a final concentration of 2 mM and the sample was incubated in the dark at 4°C for 90 min. Ethylene glycol was added to a final concentration of 10 mM to both samples (for consumption of excess periodate in the periodate treated sample) and incubated for 5 min at room temperature. NaBH$_4$ (5 M in 2.5% NaOH) was then added to both samples to a final concentration of 250 mM (pH>13) and the samples were incubated at room temperature for 1 hr. Samples were neutralized with 10% acetic acid and then concentrated and buffer exchanged into 40 mM $NH_4HCO_3$ using EMD Millipore 10 kDa NMWL Microcon centrifugal filters (final ratio 1:100). Flow-through was saved for glycan analysis. Aliquots of buffer exchanged protein were saved for dot blot analysis.

## Glycosidase treatment

Buffer exchanged protein samples as well as C18 purified tryptic peptides were treated with the GlycoPro Enzymatic Deglycosylation Kit along with the prO-LINK Extender from Prozyme (http://prozyme.com/products/gk80115) in 100 mM MES pH 6.5. Sialidase A, O-Glycanase, β (*Ervasti et al., 1990*; *Yoshida and Ozawa, 1990*; *Carmignac and Durbeej, 2012*; *Dong et al., 2015*) Galactosidase and β-N-acetylglucosamindase were applied simultaneously and incubated for six hours at 37°C.

## Trypsin digestion

Prior to glycosidase treatment, samples for MS analysis were digested according to standard protocols (*Fakhouri et al., 2006*; *Stalnaker et al., 2010*).

## Dot blots

α-DG-340 and processed samples were analyzed by dot blot on PVDF membranes. IIH6C4 antibody (Millipore) was used as primary. Dot blots were carried out with serial dilutions and secondary controls alone (not shown).

## Glycopeptide LC-MSn and data analysis

Glycopeptides were analyzed on Orbitrap Fusion and Orbitrap Fusion Lumos instruments with liquid chromatography carried out using an Acclaim PepMap RSLC C18 2 μm particle 15cm column on an Ultimate 3000 (Thermo/Dionex). Buffer A was 0.1% formic acid, buffer B was 80% acetonitrile and 0.1% formic acid. The column was heated to 45°C, equilibrated for 5 min, and a linear gradient from 5%B to 45%B was run over the course of 120 min with a 300 μl/min flow rate. The column was cleaned after each run by ramping to 99%B for 10 min and then returning to 5%B to re-equilibrate. Spray was via a stainless steel emitter with spray voltage set to 2200 V, ion transfer tube temperature 280°C. MS methods consisted of full MS scans in the Orbitrap, generally from 500–1700 m/z with quadrupole isolation. Peptide MIPs, charge state selection allowing for states 2–7, dynamic exclusion for 30 s after 2 selections with tolerance of 15 ppm on each side, and precursor priority of MostIntense were used. All MS2 and MS3 scans were analyzed in the ion trap. One branch consisted of CID scans with pseudo-loss triggered CID and HCD for phosphate, hexose and hexnac combinations with either no charge loss or a charge loss of +1. A second branch consisted of an HCD node leading to product ion triggered ETD given the observation of at least 3 oxonium ions generated by either hexose or hexnac with at least 20% relative intensity. ETD nodes were typically set for reaction times of 100–200 ms, ETD reagent targets of 2e5 or 4e5 and supplemental activation (primarily EThcD) depending on m/z. Higher m/z ions were subjected to to 40%–45% supplemental activation to disrupt low charge density clusters. Data was analyzed with Preview and Byonic versions 2.6 and 2.7 as well as by manual interpretation using Thermo Xcalibur.

## Expression and purification of human ISPD

The DNA coding sequence for human ISPD (Residues 43–451, Uniprot A4D126) was generated by gene synthesis (Life Technologies, ThermoFisher Scientific) with sequences appended to the NH$_2$-terminus comprised of a Kozak sequence followed by an initiating methionine, an 8×His-tag and a TEV-protease cleavage site and the ISPD synthetic gene followed by a termination codon at the end of the ISPD coding region. The resulting sequence was subcloned into the pGEc2 vector as described for the α-DG340-pGEc2 construct except the presence of the termination codon precluded the inclusion of the vector encoded COOH-terminal fusion sequences. Suspension culture HEK293 cells (FreeStyle 293-F cells, ThermoFisher Scientific) were transiently transfected with the ISPD-pGEc2 plasmid as described for the α-DG340-pGEc2 construct. Cells were harvested by centrifugation at 1000 x g for 10 min, 5 days post-transfection. The cell pellet was resuspended in lysis buffer [25 mM HEPES-NaOH pH 7.2, 400 mM NaCl, 20 mM imidazole, 0.3% Triton X-100, and 1× Protease Inhibitor Cocktail Set V, EDTA-Free (Calbiochem)] on ice, lysed by probe sonication for 3 cycles (15 s on, 15 s off at 40% intensity), and the cell lysate was centrifuged at 18,000 x g at 4°C for 30 min. The supernatant was subjected to Ni-NTA chromatography, and His8-hISPD was eluted using 300 mM imidazole. Fractions containing His8-hISPD were pooled, buffer exchanged into Tris-buffered saline (TBS, pH 8.0) and concentrated to 0.3 mg/mL using an Amicon Ultra-15 Centrifugal Filter Unit equipped with a 30,000 NMWL (nominal molecular weight limit) membrane (EMD Millipore).

## Expression and purification of human dTM-TMEM5

The DNA coding sequence for human transmembrane deleted dTM-TMEM5 (residues 33–443, Uniprot Q9Y2B1) was generated by gene synthesis (Life Technologies, ThermoFisher Scientific) and subcloned into a mammalian expression vector (pGEn2) containing an amino-terminal signal sequence, 8×His-tag, AviTag, and 'superfolder' GFP followed by a TEV-protease cleavage site (*Meng et al., 2013*). Suspension culture HEK293 cells (FreeStyle 293-F cells, Invitrogen) were transiently transfected with the TMEM5-pGEn2 plasmid to generate a soluble secreted GFP-dTM-hTMEM5 and the protein was harvested, purified, and concentrated as described for α-DG340.

## TMEM5 sugar-nucleotide specificity assay

Ultra Pure UDP-Glc, UDP-GlcNAc, UDP-Gal, UDP-GalNAc, and UDP-GlcA, were purchased from Promega. Ultra Pure UDP-Xyl was prepared by incubating 10 µmol UDP-Xylose (Carbosource, University of Georgia) with 3 units of Calf Intestinal Alkaline Phosphatase (CIAP, Promega) in $1\times$ CIAP Buffer (50 mM Tris-HCl pH 9.3, 1 mM $MgCl_2$, 0.1 mM $ZnCl_2$ and 1 mM spermidine) at 37°C for 16 hr. CIAP was used to degrade any contaminating nucleotide diphosphates that may contribute to background levels in the downstream UDP-Glo Glycosyltransferase Assay (Promega). Reactions were allowed to incubate for 16 hr and were stopped by removal of CIAP by filter centrifugation through a Microcon-10 kDa Centrifugal Filter Unit with Ultracel-10 membrane (EMD Millipore).

Specificity of GFP-hTMEM5 sugar-nucleotide hydrolysis was performed by incubation of 3 µM GFP-hTMEM5 with 50 µM of UDP-sugar (Ultra Pure UDP-Glc, UDP-GlcNAc, UDP-Gal, UDP-GalNAc, UDP-GlcA, or UDP-Xyl) in the absence of an acceptor substrate in a 20 µL reaction containing 0.1 M MES pH 6.0 and 10 mM $MgCl_2$ at 37°C for 18 hr. Detection of free UDP after hydrolysis of the sugar-nucleotide was performed using the UDP-Glo Glycosyltransferase Assay Kit (Promega) which detects released UDP by converting UDP to ATP and then light in a luciferase reaction, which can be measured by a luminometer. Luminescence detected is directly proportional to UDP concentration, as determined by a UDP standard curve from 0 to 25 µM. Essentially, each sugar-nucleotide hydrolysis reaction was combined in a ratio of 1:1 (5 µL:5 µL) with the UDP-Glo Detection Reagent from the assay kit in separate wells of a white, flat bottom 384-well assay plate (Corning) and allowed to incubate at room temperature for 1 hr. Luminescence was measured in triplicate using a Promega GloMax-Multi+ Microplate Luminometer.

## Preparation of CDP-ribitol and CDP-ribose

Ribitol (adonitol), D-ribose, ribose-5-phosphate, and cytidine 5'triphosphate (CTP) were purchased from Sigma-Aldrich. Ribitol-5-phosphate was prepared essentially as previously described (*Baddiley et al., 1956*). Confirmation of ribitol-5-phosphate was performed by 1D NMR, in addition to observing the complete disappearance of ribose-5-phosphate anomeric signals after reduction by sodium borohydride. Quantification of ribitol-5-phosphate was performed by peak integration of the 1H signals in the 1D 1H NMR spectrum compared to the internal standard 2,2-dimethyl-2-silapentane-5-sulfonate (DSS) at 0.00 ppm (data not shown).

## TMEM5 glycosyltransferase assay

GFP-dTM-hTMEM5 (from 0 to 2500 nM) was incubated with 1 mM acceptor substrate and 50 µM UDP-Xyl donor substrate at 37°C in a reaction containing 0.1 M MES pH 6.0 and 10 mM $MgCl_2$ for 18 hr. Detection of free UDP after hydrolysis of the sugar-nucleotide was performed using the UDP-Glo Glycosyltransferase Assay (Promega) as described above. CMP-Neu5Ac was purchased from Sigma-Aldrich.

## Measuring pyrophosphorylase activity

To prepare cytidine diphosphate ribitol (CDP-ribitol) and cytidine diphosphate ribose (CDP-ribose), 2 µM His8-hISPD was incubated at 37°C with either 1 mM ribitol-5-phosphate or 1 mM ribose-5-phoshate as the acceptor substrate in a reaction containing 50 mM Tris-HCl pH 7.4, 1 mM $MgCl_2$, 1 mM DTT, and 1 mM CTP. Reactions were allowed to incubate for 16–18 hr and were stopped by removal of His8-hISPD by filter centrifugation through a Microcon-10kDa Centrifugal Filter Unit with Ultracel-10 membrane (EMD Millipore). Reaction products were confirmed using a linear ion trap-Fourier transform mass spectrometer (LTQ-Orbitrap Discovery, Thermo-Fisher, San Jose, CA). Reaction products were mixed with an equal volume of 80% acetonitrile and 0.1% formic acid and analyzed by direct infusion in negative ion mode using a nanospray ion source with a fused-silica emitter (360 $\times$ 75 $\times$ 30 µm, SilicaTip, New Objective) at 1.5 kV capillary voltage, 200°C capillary temperature, and a syringe flow rate of 1 µL/min. All products were confirmed by MS/MS ion trap mass spectrometry (ITMS) acquired at 45% collision-induced dissociation (CID) and 2 m/z isolation width.

## DGFc4 glycosidase and IIH6 assays

5µg aliquots of DGFc4 previously purified from the media of HEK293H cells co-transfected with LARGE (*Yoshida-Moriguchi et al., 2010*) was incubated overnight with each of the following

glycosidases: chondroitinase A,B,C, sialidase A, heparinase I, heparinase II, and β-N-acetylhexosaminidase. Additionally, 5 μg of DGFc4 was treated with HF as reported previously (*Yoshida-Moriguchi et al., 2010*). All samples were later separated by SDS-PAGE, transferred, and immunoblotted with IIH6 for assessment of functionally active α-DG.

## Cell cultures

Cells were maintained at 37°C and 5% $CO_2$ in Dulbecco's modified Eagle's medium (DMEM) plus fetal bovine serum (FBS: 10% in the case of HEK293T cells, 20% in the case of fibroblasts from patient skin) and 2 mM glutamine, 0.5% penicillin-streptomycin (Invitrogen, Carlsbad, CA). Mycoplasma free conditions were verified by PCR and cell lines used are not on the registry of commonly misidentified cells.

## Cloning of C-terminal HA-tagged TMEM5 wildtype and TMEM5 p. G333R mutant constructs

The human *TMEM5* coding sequence was PCR amplified and a C-terminal HA epitope-tag was introduced with PCR adapters using the following primer sequences:

h*TMEM5*-HA *wildtype* (1.3 kb), pTW292: forward (5'-aga**ctcgag**accATGcggctgacgcggaagcg-3', where the XhoI adapter is bolded and the start ATG codon is shown in capital letters) and reverse (5'- ctt**gcggccgc**CTAAGCGTAGTCTGGGACGTCGTATGGGTAgctagccccactttttattattcattaaaaatg-3'; the NotI adapter is bold and the HA-tag is shown in capital letters). The h*TMEM5*-HA PCR fragment (*TMEM5*, NM_014254) was subloned in pIRES-hygromycin.

h*TMEM5*-HA mutant (c.997 G>A, p.G333R) (1.3 kb), pTW293: To generate the human *TMEM5*-HA p.G333R mutant expression construct, we introduced the missense mutation in the wildtype template pTW292 using a QuikChange site-directed mutagenesis kit (Agilent Technologies, Santa Clara, CA) with overlapping primers that included the respective mutation: forward 5'-gtgcccggtc**A**gagtaaacacagaatg-3' and reverse: 5'-gtgtttactc**T**gaccgggcacaatgtg -3'. The introduced mutation is highlighted (capital bold letter). The sequence of the insert DNA was confirmed by Sanger sequencing.

## Adenovirus generation and gene transfer

E1-deficient recombinant adenoviruses (Ad5CMV-α-DG-Fc340, Ad5CMV-α-DG-Fc340 mut (TPT, 317–319 to APA) and Ad5CMV-*LARGE*/RSVeGFP) were generated by the University of Iowa Gene Transfer Vector Core and have been described previously (*Barresi et al., 2004*). The constructs used to generate the E1-deficient recombinant adenoviruses Ad5CMV-α-DG-Fc340 and Ad5CMV-α-DG-Fc340 mut (TPT, 317–319 to APA) were made from pcDNA3-α-DG-Fc340 and pcDNA3-α-DG-Fc340 mut (TPT, 317–319 to APA) (*Hara et al., 2011*). pcDNA3-α-DG-Fc340 vectors were digested with KpnI/XbaI, and the resulting fragments were ligated into a KpnI/XbaI-digested pacAd5-CMV-NpA vector. To generate Ad5CMV-*TMEM5*-Myc/RSVeGFP we cloned a 1.3 kb XhoI/SpeI h*TMEM5*-Myc fragment corresponding to human *TMEM5* (*TMEM5*, NM_014254) from pTW170 into the XhoI/SpeI polylinker region of pAd5CMVK-NpA (pTW174). Cultured cells were infected with viral vectors for 12 hr, at an MOI of 200. We examined cultures 3–5 d after treatment.

We also used nucleofection as a non-viral method for gene transfer into cells. Nucleofection of fibroblasts was performed using the Human Dermal Fibroblast Nucleofector Kit, according to an optimized protocol provided by the manufacturer (Amaxa Biosystems, Germany).

## Glycoprotein enrichment and biochemical analysis

Zebrafish embryos (48 h.p.f.) were deyolked, followed by microsome preparation and western blot analysis as previously described (*Roscioli et al., 2012*; *Link et al., 2006*). Relative signal intensity of western blot was quantified using ImageJ software.

WGA-enriched glycoproteins from frozen samples and cultured cells were processed as previously described (*Michele et al., 2002*). Immunoblotting was carried out on polyvinylidene difluoride (PVDF) membranes as previously described (*Michele et al., 2002*). Blots were developed with IR-conjugated secondary antibodies (Pierce Biotechnology, Rockford, IL) and scanned with an Odyssey infrared imaging system (LI-COR Bioscience, Lincoln, NE).

The monoclonal antibodies to the fully glycosylated form of α-DG (IIH6) (*Ervasti and Campbell, 1991*), and also the polyclonal antibodies rabbit β-dystroglycan (AP83) (*Duclos et al., 1998*) were characterized previously. G6317 (core-α-DG) from rabbit antiserum was raised against a keyhole limpet hemocyanin (KLH)-conjugated synthetic peptide of human dystroglycan (*Willer et al., 2012*). Mouse monoclonal anti-c-Myc (clone 4A6) antibodies were purchased from Millipore (Billerica, MA).

## Laminin overlay assays
Laminin overlay assays were performed on PVDF membranes using standard protocols (*Michele et al., 2002*).

## Immunofluorescence microscopy
HEK293T cells were transfected with *TMEM5*-HA constructs using FuGENE 6 (Roche Applied Science, Penzberg, Germany). After 48 hr cells were fixed with 4% paraformaldehyde in PBS and permeabilized with 0.2% Triton X-100 in PBS for 10 min on ice. After blocking with 3% BSA in PBS, the slides were incubated with monoclonal anti-HA antibody (Clone 16B12, COVANCE, Emeryville, CA) rabbi-polyclonal anti-Giantin (ab24586, Abcam, Cambridge, MA) for 18 hr at 4°C. The cells were incubated with an appropriate secondary antibody conjugated to AlexaFluor488 or AlexaFluor555 fluorophore after washing with PBS. 4',6'-Diamidino-2-phenylindole dihydrochloride (DAPI, Sigma, St Louis, MO) was used for nuclear staining. Cryosections (10 µm thick) of the fetal skeletal muscle sample were processed for immunofluorescence as described (*Duclos et al., 1998*). Mouse monoclonal antibodies included anti-alpha-dystroglycan (IIH6 and VIA4-1; Millipore, Billerica, MA), anti-beta-dystroglycan (7D11; DSHB), anti-laminin α2 (anti-merosin, 5H2; Millipore, Billerica, MA), and anti-spectrin (NCL-SPEC1; Leica, Buffalo Grove, IL). A rabbit polyclonal anti-dystrophin C-terminus antibody was from Abcam (ab15278) and a goat polyclonal anti-alpha-dystroglycan core antibody (Gt20) (*Michele et al., 2002*) were used. Secondary antibodies were conjugated with AlexaFluor488. Nuclei were stained with DAPI by using ProLong Gold mounting medium (Molecular Probes/ThermoFisher, Waltham, MA). Images were obtained using a Zeiss Zeiss LSM710 confocal microscopy (Carl Zeiss, Thornwood, NY).

## On-Cell complementation and western blot assay
The On-Cell complementation assay was performed as described previously (*Willer et al., 2012*). In brief, $2 \times 10^5$ cells were seeded into a 48-well dish. The next day the cells were co-infected with 200 MOI of Ad5RSV-DAG1 (*Barresi et al., 2004*) and Ad5CMV-*TMEM5*-Myc/RSVeGFP in growth medium. Three days later, the cells were washed in TBS and fixed with 4% paraformaldehyde in TBS for 10 min. After blocking with 3% dry milk in TBS + 0.1% Tween (TBS-T), the cells were incubated with primary antibody (glyco α-DG, IIH6) in blocking buffer overnight. To develop the On-Cell Western blots we conjugated goat anti-mouse IgM (Millipore, Billerica, MA) with IR800CW dye (LI-COR Bioscience, Lincoln, NE), subjected the sample to gel filtration, and isolated the labeled antibody fraction. After staining with IR800CW secondary antibody in blocking buffer, we washed the cells in TBS and scanned the 48-well plate using an Odyssey infrared imaging system (LI-COR Bioscience, Lincoln, NE). For cell normalization, DRAQ5 cell DNA dye (Biostatus Limited, United Kingdom) was added to the secondary antibody solution.

## TMEM5 [Xyl-$^{14}$C] radioactive sugar donor in vitro assay
To generate α-DG-Fc340 and α-DG-Fc340-mut (TPT, 317–319 to APA) acceptor proteins, we infected control and glycosylation-deficient *TMEM5*-WWS patient skin fibroblasts with Ad5-CMV α-DG-Fc340 adenoviral vectors at an MOI of 400. At 4 days post-infection the secreted proteins were isolated from the culture medium using Protein A-agarose beads (Santa Cruz, Dallas, TX). α-DG-Fc340 bound Protein A-agarose beads were washed three times with TBS and Protein A slurry pre-bound with ~25 µg α-DG-Fc340 was added to the in vitro TMEM5 assay. Enzyme reactions (25 µl) were carried out at 37°C for 6 hr, with 0.1 µCi UDP-Xyl [Xyl-$^{14}$C] (final conc. 15.8 µM), in 0.1 M MOPS buffer (pH 6.5) supplemented with 10 mM MnCl$_2$, 10 mM MgCl$_2$, 0.2% BSA and 1 µg purified TMEM5 protein (Origene, Rockville, MD). The reaction was terminated by adding 25 µl of 0.1 M EDTA. After four washes with TBS the Protein A-agarose-bound α-DG-Fc340 samples were analyzed

by scintillation counting. [$^{14}$C] labeled sugar nucleotides were purchased from ARC (American Radio-labeled Chemicals, St. Louis, MO).

## Antisense morpholino oligonucleotides (MO)

An antisense *tmem5* MO targeting the translation start site was designed and ordered from Gene Tools (Philomath, OR). The sequence of *tmem5* MO is: 5'-CCGGCGAAAAAATCT(CAT)<u>GTTGGAT</u>-3' (start codon in brackets with the 5'-UTR sequence underlined). Sequences of *p53* and *dag1* MOs have been described (*Robu et al., 2007*; *Parsons et al., 2002*). MOs were injected into zebrafish embryos by the 2-cell stage with concentration specified in the figures.

## Molecular cloning of zebrafish *tmem5* and mRNA synthesis

Full length *tmem5* coding sequence was amplified from IMAGE cDNA clone (7450642) using PCR primers to obtain a PCR product including 7 bases before start codon, yet excluding the stop codon. The PCR product was subsequently cloned into pCS2+_EGFP expression vector using Gateway clonase system (Invitrogen, Carlsbad, CA). Sense *tmem5-egfp* mRNA containing full *tmem5* MO binding site was synthesized using mMESSAGE mMACHINE SP6 kit (Ambion, Austin, TX).

## Reverse transcription, cDNA synthesis and PCR

Wildtype or MO-injected Zebrafish embryos were collected at specific stages and homogenized to extract total RNA using TRIzol (Invitrogen, Carlsbad, CA). First-strand cDNA was synthesized using SuperScript III (Invitrogen, Carlsbad, CA) with either oligo dT or random primers.

## Statistical analysis in zebrafish

Eye width measurements were plotted as mean ± s.d. and statistical significance was determined using one-way ANOVA, followed by Tukey HSD test. A p value smaller than 0.01 was considered statistically significant.

## Evans blue dye (EBD) injection

0.1% EBD (Sigma) was injected into the blood circulation of zebrafish embryos at least 1 hr before analysis. The sarcolemma integrity was then assessed using confocal and differential interference contrast (DIC) microscopy at 48 h.p.f.

## Human subjects and samples

All tissues and patient cells were obtained and tested according to the guidelines set out by the Human Subjects Institutional Review Board of the University of Iowa; informed consent was obtained from all subjects or their legal guardians.

## Genotyping and IBD/HBD analysis

High molecular weight genomic DNA samples from three cases were genotyped on Illumina Omni-1 Quad BeadChip at the Southern California Genotyping Consortium (SCGC, http://scgc.genetics.ucla.edu/) or Affymetrix Human Mapping 250K NSP at the UCLA Genome Sequencing Center (http://gsc.ucla.edu/). Homozygosity-by-descent (HBD) analysis was performed using a custom Mathematica script available at Wolfram Research; B. Merriman, http://genome.ucla.edu/~hlee/script_public/HBD_IBD/HBD_IBD_Script.nb and the interval file used is available as Source code. The HBD analysis simply searches for long stretches of homozygous calls within each individual (*Lee et al., 2008*). A conservative error rate of 1% was used to allow the algorithm to tolerate possible genotyping errors. Intervals over 10 cM with stretches of homozygous genotypes are indications that the individual is a descendent of a consanguineous marriage. To evaluate the sharing between the siblings, SNP positions where the genotypes were not only homozygous in each individual but also the same allele were noted as shared.

## Capture array design and targeted sequencing

Regions over 3 cM that were homozygous in each sample were identified and pairwise comparisons were performed to find subset of the regions where the two affected individuals, TR and 02243-a, were homozygous with the same allele and the unaffected individual, 02243-b, was heterozygous or

homozygous with an alternate allele. All coding exon regions within those regions were retrieved using six different exon/gene prediction models (RefSeq (refGene), UCSC (knownGene), Vertebrate Genome Annotation genes (vegaGene), Ensembl genes (ensGene), consensus coding sequence (ccdsGene), and Mammalian Gene Collection genes (mgcGene). In total, 6647 coding exons plus the regions extending 10 bp on each side of each exon across 4 intervals were subject to capture probe design using Agilent eArray (www.agilent.com). Agilent custom CGH array platform was chosen, requiring the probes to have the melting temperature (Tm) around 80°C (default) with probe trimming allowed. Repeat regions were excluded for probe design by turning on the 'Avoid Standard Repeat Masked Regions' option.

Genomic DNA was extracted from skin fibroblasts using Qiagen (Hilden, Germany) DNeasy Blood & Tissue Kit was run on Qubit Fluorometer (Invitrogen) and Bioanalyzer (Agilent) for quality check. For each sample, 3 µg of high molecular weight genomic DNA was used as starting material, the sequencing library was prepared following Agilent SureSelect SureSelect Target Enrichment System for Illumina Paired-End Sequencing Library Protocol (version 2.0.1). Instead of using the commercial adapter, 3 different custom made bar-coded adapters were used; TR: Forward: ACA CTC TTT CCC TAC ACG ACG CTC TTC CGA TCT TGA GT

Reverse: /5Phos/CTC AAG ATC GGA AGA GCG GTT CAG CAG GAA TGC CGA G
02243-a: Forward: ACA CTC TTT CCC TAC ACG ACG CTC TTC CGA TCT CAT CT
Reverse: /5Phos/GAT GAG ATC GGA AGA GCG GTT CAG CAG GAA TGC CGA G
02243-b: Forward: ACA CTC TTT CCC TAC ACG ACG CTC TTC CGA TCT AAT AT
Reverse: /5Phos/TAT TAG ATC GGA AGA GCG GTT CAG CAG GAA TGC CGA G. After amplification, samples were pooled at equal molar concentration, captured on one array following an in-house protocol (*Lee et al., 2009*) and sequenced on approximately 3/11th lane of Illumina HiSeq2000 as 50bp paired-end reads, following the manufacturer's protocol. The base-calling was performed by the real time analysis (RTA) software provided by Illumina.

## Sequence read alignment

The sequence reads were first de-barcoded using Novobarcode from Novocraft Short Read Alignment Package (http://www.novocraft.com/ index.html) and aligned to the Human reference genome, human_g1k_v37.fasta using Novoalign. The reference genome downloaded from the GATK (The Genome Analysis Toolkit) resource bundle (http://www.broadinstitute.org/gsa/wiki/index.php/Main_Page) in November, 2010 was indexed using novoindex (–k 14 –s 3). The output format was set to SAM and default options for alignment were applied except for the adaptor stripping option (-a) and base quality calibration option (-k). Using SAMtools (http://samtools.sourceforge.net/) version 0.1.15, the SAM file of each sample was converted to BAM file and sorted, and potential PCR duplicates were removed (rmdup) using Picard (http://picard.sourceforge.net/). Local realignment was performed using GATK 'IndelRealigner' tool per sample. First, the 'RealignerTargetCreator' tool was used to determine the locations that are potentially in need of realignment. The post-rmdup bam file and the known SNP positions (Single Nucleotide Polymorphisms) in dbSNP132 were included as inputs and '–mismatchFraction 0.10' and '–realignReadsWithBadMates' options were used. Using the intervals created, reads were realigned and the mates were fixed using Picard's 'FixMateInformation' tool. Base qualities were recalibrated using GATK 'TableRecalibration' tool by analyzing the covariates for the reported base quality score of a base (QualityScoreCovariate), the combination of a base and the previous base (DinucCovariate) and the machine cycle for a base (CycleCovariate).

## Variant calling

Variants were called using GATK 'Unified Genotyper' tool simultaneously for all 8 samples. Small indels were called with the '-glm DINDEL' option. The dbSNP132 file downloaded from the GATK resource bundle was used so that the known SNP positions are annotated in the output VCF (variant call format) file. Variants with phred-scale Qscore of 50.0 or greater were reported as 'PASS'-ed calls and those with Qscore of 10.0 or greater and less than 50.0 were reported as 'Low Qual' calls. Variants with Qscore less than 10.0 were not reported. Only the variants found within the protein coding regions of the captured exons were reported by using the –L option. The interval file used is available as Source code. Using GATK 'VariantFiltrationWalker' tool, both the SNPs and INDELs were hard-filtered to filter out low quality variants. The following parameters were used as

suggested by GATK as standard filtration: 1) clusterWindowSize 10; 2) MAPQ0 (mapping quality of zero) >40; 3) QD (Quality-by-depth) < 5.0; 4) SB (Strand Bias) > -0.10.

## Variant annotation

The 'PASS'-ed variants that are not found at dbSNP132 positions were annotated using SeattleSeqAnnotation version 6.16 (http://snp.gs.washington.edu/SeattleSeqAnnotation131/), separately for SNPs and INDELs. Both NCBI full genes and CCDS 2010 gene models were used for the annotation. For variants with multiple annotations, if the 'functionGVS' (Genome Variation Server class of function) was different while the 'genelist' value was same, the one with protein level change was retained. Variants present in the 1000 Genomes database (March 2010 release) or dbSNP131 as well as those resulting in coding-synonymous changes or found outside the coding region were removed from further analysis.

## Sanger sequencing

Genomic DNA was extracted from dermal fibroblast cell lines using a Qiagen (Hilden, Germany) DNeasy Blood & Tissue Kit. The coding regions (6 exons) and exon-intron boundaries of *TMEM5* were amplified using PCR (Primers sequences and PCR conditions are shown below). Primer sets for PCR were designed using the web-based design tool ExonPrimer. After PCR amplification the purified products were evaluated by Sanger sequencing using standard protocols.

**h*TMEM5* patient sequencing primers**

**gDNA PCR amplification**

| primer ID | primer name | sequence | length |
|---|---|---|---|
| 7630 | TMEM5g.E1F | CTCCTCCAGTGTCGAAGGTG | 20 |
| 7642 | TMEM5g.E2R3 | TGCTCTGCTAAAGACCAACAGTGG | 24 |
| 7628 | TMEM5g.E3F | TGATTTGGAGCTGTTGCTTG | 20 |
| 7627 | TMEM5g.E3R | TGTAAGATGTGGAGTATGAACTTTTC | 26 |
| 7626 | TMEM5g.E4F | CCACATACCTTTGTTCAGGC | 20 |
| 7625 | TMEM5g.E4R | CCTGGTTCAGAAATTCAGGTG | 21 |
| 7624 | TMEM5g.E5F | GGAGTTTTCCAAAGTATTCATGG | 23 |
| 7623 | TMEM5g.E5R | ATCTTCTGGGGAAAGATTGG | 20 |
| 7622 | TMEM5g.E6F | AAGAAATCTGTTTGGGCCAG | 20 |
| 7621 | TMEM5g.E6R | TGCAATACATATGTCATCACTAGGC | 25 |

**PCR sequencing**

| primer ID | primer name | sequence | length |
|---|---|---|---|
| 7641 | TMEM5g.E1F2 | ATGAGCCGCGACTGGAGG | 18 |
| 7639 | TMEM5g.E2F2 | TTTTGTTTTCATTGTGTATTACCAG | 25 |
| 8333 | hTMEM5_E3F2 | TGTTGCTTGATAGCACTGCCTG | 22 |
| 8334 | hTMEM5_E4F2 | TGATGAATTTCTGATACCACATACC | 25 |
| 8335 | hTMEM5_E5F2 | TAAGAGATTGGGTTATGGGG | 20 |
| 8336 | hTMEM5_E6F2 | CCAGGATTTTGGATATCTCTTATG | 24 |

**Patient sequencing: hTMEM5**

**PCR amplify gDNA fragments**

| primer | exons | PCR fragment size in bp | sequencing primer | |
|---|---|---|---|---|
| 7630/7642 | exon1-2 | 1682 | 7641 | exon1 |
| | | | 7639 | exon2 |
| 7628/7627 | exon3 | 547 | 8333 | exon3 |
| 7626/7625 | exon4 | 433 | 8334 | exon4 |
| 7624/7623 | exon5 | 453 | 8335 | exon5 |
| 7622/7621 | exon6 | 833 | 8336 | exon6 |

**PCR-Program**

| | | Temp. | Duration |
|---|---|---|---|
| 1) Denaturation | | 95 | 3 min |
| 2) Cycles | | 95 | 30 s |
| 3) Cycles | (35x) | 53 | 30 s |
| 4) Cycles | | 72 | 1 min 30 s |
| 5) Elongation | | 72 | 10 min |

## Acknowledgements

We would like to thank all members of the Wells, Campbell, and Moremen laboratories for helpful discussions. We are indebted to Dr. Michael Tiemeyer for valuable discussions regarding experiments presented here. We would like to thank Allison Schwartz and Traci Toy at the UCLA Genome Sequencing Center (http://gsc.ucla.edu/) for assisting with genotyping and constructing the sequencing libraries and Dr. Suhua Feng at the UCLA Broad Stem Cell Research Center (BSCRC) for assisting in running HiSeq2000. We thank Greg Morgensen and David Venzke for technical support and Christine Blaumueller for critical reading of the manuscript. This work was supported in part by grants from NIGMS/NIH (R01GM111939 to LW, P01GM107012, KWM and LW co-PIs), technology resource grants from NIGMS/NIH (P41GM103490, LW and KWM co-PIs and P41GM103390, KWM, PI), a Paul D. Wellstone Muscular Dystrophy Cooperative Research Center Grant (1U54NS053672, KPC, SAM and TW), a MDA grant (238219, KPC and TW) and an ARRA Go Grant (1 RC2 NS069521-01, KPC and TW). KPC is an investigator of the Howard Hughes Medical Institute.

## Additional information

### Funding

| Funder | Grant reference number | Author |
|---|---|---|
| National Institute of Neurological Disorders and Stroke | U54NS053672 | Tobias Willer<br>Steven A Moore<br>Kevin P Campbell |
| Muscular Dystrophy Association | 238219 | Tobias Willer<br>Kevin P Campbell |
| National Institute of General Medical Sciences | R01GM111939 | Kelley W Moremen<br>Lance Wells |
| National Institute of General Medical Sciences | P41GM103390 | Kelley W Moremen<br>Lance Wells |
| Howard Hughes Medical Institute | | Kevin P Campbell |
| National Institute of General Medical Sciences | P01GM107012 | Kelley W Moremen<br>Lance Wells |
| National Institute of General Medical Sciences | P41GM103490 | Kelley W Moremen<br>Lance Wells |
| National Institute of Neurological Disorders and Stroke | RC2 NS069521-01 | Kevin P Campbell<br>Tobias Willer |

The funders had no role in study design, data collection and interpretation, or the decision to submit the work for publication.

### Author contributions

JLP, TW, MOS, Co-designed the project, Carried out experimental work, Analyzed and interpreted the data, Co-wrote the manuscript; AT, DC, Provided clinical data and patient samples, Acquisition of data, Analysis and interpretation of data; Y-YL, Co-wrote the manuscript, Performed zebrafish

experiments, Conception and design, Acquisition of data, Analysis and interpretation of data; HL, SFN, Performed next generation sequencing and data filtering, Acquisition of data, Analysis and interpretation of data; SHS, SW, Carried out experimental work, Analyzed and interpreted the data; PKP, Carried out experimental work, Analyzed and interpreted the data, Acquisition of data, Analysis and interpretation of data; DLS, Supervised zebrafish experiments, Conception and design, Analysis and interpretation of data; SAM, Performed muscle histology and clinical data interpretation, Co-wrote the manuscript; KWM, Designed protein expression, Analyzed and interpreted the data, Co-wrote the manuscript, Supervised the protein expression research; KPC, LW, Co-designed the project, Analyzed and interpreted the data, Co-wrote the manuscript, Supervised the research

### Author ORCIDs
M Osman Sheikh, http://orcid.org/0000-0002-9481-8318
Yung-Yao Lin, http://orcid.org/0000-0002-0435-7694
Kevin P Campbell, http://orcid.org/0000-0003-2066-5889
Lance Wells, http://orcid.org/0000-0003-4956-5363

### Ethics
Human subjects: Informed consent and ethical approval is detailed in the Materials and Methods section. All tissues and patient cells were obtained and tested according to the guidelines set out by the Human Subjects Institutional Review Board of the University of Iowa; informed consent was obtained from all subjects or their legal guardians.

## Additional files

### Supplementary files
• Source code 1. HBD analysis.

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
