## [Decision Letter]

Thank you for submitting your work entitled "The Functional O-Mannose Glycan on α-Dystroglycan is a Trisaccharide-phospho-ribitol Primed for Matriglycan Addition" for consideration by *eLife*. Your article has been favorably evaluated by Ivan Dikic as Senior editor and three reviewers, one of whom, Amy Wagers, is a member of our Board of Reviewing Editors.

The reviewers have discussed the reviews with one another and the Reviewing Editor has drafted this decision to help you prepare a revised submission.

Summary:

This manuscript describes new insights into a glycan on α-dystroglycan (α-DG) required for the binding of ECM ligands like laminin. Previous work from the Campbell and Wells labs has reported structural characterization of the O-mannose glycans on the mucin domain of α-DG, and the glycan generated by LARGE, termed matriglycan. The experiments described here propose that ribitol-Xyl-GlcA is the link from the phosphate on O-Mannose to matriglycan. The authors also propose that two genes implicated in CMDs, ISPD and TMEM5, encode a CDP-ribitol pyrophosphorylase and a xylosyltransferase, respectively. They also identify a new mutation in TMEM5 that may be the basis of WWS in a consanguineous family. The work has many strengths, including a comprehensive approach that combines biochemical and structural evaluations, use of in vivo vertebrate (zebrafish) models, and genetic characterization and phenotypic rescue studies in human WWS patients and patient-derived cells, and overall, represents an extremely high quality contribution to our understanding of the assembly of the glycans that modify α-dystroglycan, and how these enable binding of α-dystroglycan to ECM proteins. However, there are also a number of 'loose ends', some of which are outlined at the end of the Discussion. Unfortunately, these preclude many of the definitive statements made throughout the manuscript. These issues should be addressed through the essential revisions outlined below.

Essential revisions:

1) Figure 2 and Figure 2—figure supplement 1 show MS and MS/MS spectra that are annotated with glycans for which there is no evidence except m/z. The linkages must be removed from the proposed structures. Are any other sugar combinations consistent with the data? If so, they should be presented. Since no data in the manuscript addresses linkage, linkages in the proposed structure should be removed from every figure.

2) TMEM5 hydrolyzes UDP-Xyl but not UDP-GlcNAc or UDP-GlcA. However, UDP-GalNAc, UDP-Gal and UDP-Glc should also be tested since conclusions re structures were made solely from m/z.

3) The authors show that purified hTMEM5 transfers label from UDP-^14^C-Xyl to α-DG but not to α-DG APA from patient's cells lacking TMEM5. However, no product characterization was performed. This is important because very small numbers of DPM were obtained and the ^14^C-Xyl could have been transferred to something other than ribitol on the secreted α-DG substrate. The authors should treat the product with periodate and show that ^14^C-Xyl is released. They should also show that if unlabeled Xyl is transferred from UDP-Xyl, it precludes the subsequent addition of ^14^C-Xyl from UDP-^14^C-Xyl. Finally, they should show that mutant *TMEM5* G333R has reduced UDP-Xyl transferase activity. Transferase activity should be given as specific activity, and the assay more completely described in Methods, so it could be reproduced.

4) The fact that TMEM5 corrects patient cells with a TMEM5 mutation does not in itself prove that TMEM5 is the complementing gene. The authors have previously shown that overexpression of LARGE complements several different non-LARGE genes that cause CMD. *TMEM5* G333R should be compared to wild type in the complementation assay. This is an easy experiment because the authors have done this for the WT TMEM5, and makes one wonder why it was not done with the *TMEM5* G333R mutant.

5) The authors show that purified ISPD can form CDP-ribose or CDP-ribitol from CTP and ribose-P or ribitol-P. They note in the Discussion that these results were published in 2015 by another group. They should mention the published results up front in the Results section and clarify that the relationship of their approaches to this other report, including those that confirm the prior study.

6) Several abbreviations are used in the Abstract without definition (ISPD, B4GAT, LARGE, ECM). A non-expert reader likely will not understand much of this abstract, and may decide to not to read the paper (which would be too bad, since the paper is outstanding). As much as possible, these abbreviations should be defined and/or eliminated in favor of using the full word(s).

7) Throughout the manuscript, the authors need to base their conclusions and their statements firmly on their empirical data. The present text should be extensively revised to eliminate overstatements and definitive conclusions. The authors have obtained nice data that in sum are consistent with their proposed structure, but they have presented the data in many places as definitive, and even included linkages for which no data are provided. The text should make the tentative nature of their conclusions abundantly clear. Final proof will only come when the transfer of ribitol, xylose, GlcA and matriglycan can be demonstrated in vitro, using purified enzymes, and fully characterized substrates and products. Until then, statements such as "Here we elucidate the structure" in the Abstract and "Here we define a fully functional glycan structure" are unacceptable. There remain many possibilities for the roles of FKTN and FKRP as well as other mysteries to explain, including why MEB disease arises from mutation of POMGNT1, which is apparently not part of the M3 core glycan.

8) The authors should be sure to report numbers of experimental replicates and n values for *all* of their figures; some (e.g. Figure 6) show single examples, and so it is important to indicate how many times such outcomes were observed.

---

## [Author Response]

Essential revisions:

1) Figure 2 and Figure 2—figure supplement 1 show MS and MS/MS spectra that are annotated with glycans for which there is no evidence except m/z. The linkages must be removed from the proposed structures. Are any other sugar combinations consistent with the data? If so, they should be presented. Since no data in the manuscript addresses linkage, linkages in the proposed structure should be removed from every figure.

We have improved upon the presentation of this data but agree that linkage should be omitted when not specifically addressed and have done so throughout.

2) TMEM5 hydrolyzes UDP-Xyl but not UDP-GlcNAc or UDP-GlcA. However, UDP-GalNAc, UDP-Gal and UDP-Glc should also be tested since conclusions re structures were made solely from m/z.

In order to identify the sugar-nucleotide donor specificity of TMEM5, we have tested a full panel of UDP-sugars (including UDP-GalNAc, UDP-Gal and UDP-Glc). Our results demonstrate that TMEM5 can hydrolyze UDP-Xyl, but not other UDP-sugars, in the absence of an acceptor substrate. We have included these results in the updated Figure 5.

*3) The authors show that purified hTMEM5 transfers label from UDP-^14^C-Xyl to α-DG but not to α-DG APA from patient's cells lacking TMEM5. However, no product characterization was performed. This is important because very small numbers of DPM were obtained and the ^14^C-Xyl could have been transferred to something other than ribitol on the secreted α-DG substrate. The authors should treat the product with periodate and show that ^14^C-Xyl is released. They should also show that if unlabeled Xyl is transferred from UDP-Xyl, it precludes the subsequent addition of ^14^C-Xyl from UDP-^14^C-Xyl. Finally, they should show that mutant TMEM5 G333R has reduced UDP-Xyl transferase activity. Transferase activity should be given as specific activity, and the assay more completely described in the Methods, so it could be reproduced.*

While we did not pursue characterization of the radiolabeled product, we did expand upon our work in Figure 5 to include transfer of Xyl from UDP-Xyl as a purified enzyme. We were able to show that TMEM5 does not transfer efficiently in vitro to ribitol or ribitol-5-P but does transfer to CDP-ribitol suggesting that the phosphate of the acceptor likely needs to be a phosphodiester linkage. We also address the mutant in new complementation data discussed in point #4 below.

*4) The fact that TMEM5 corrects patient cells with a TMEM5 mutation does not in itself prove that TMEM5 is the complementing gene. The authors have previously shown that overexpression of LARGE complements several different non-LARGE genes that cause CMD. TMEM5 G333R should be compared to wild type in the complementation assay. This is an easy experiment because the authors have done this for the WT TMEM5, and makes one wonder why it was not done with the TMEM5 G333R mutant.*

We have performed complementation assays in patient fibroblasts with WT TMEM5 and demonstrated that *TMEM5*-G333R is not able to complement (see Figure 7).

5) The authors show that purified ISPD can form CDP-ribose or CDP-ribitol from CTP and ribose-P or ribitol-P. They note in the Discussion that these results were published in 2015 by another group. They should mention the published results up front in the Results section and clarify that the relationship of their approaches to this other report, including those that confirm the prior study.

In addition to the Discussion we now cite the manuscript in the Results section as well and clarify that both their and our data are in complete agreement.

6) Several abbreviations are used in the Abstract without definition (ISPD, B4GAT, LARGE, ECM. A non-expert reader likely will not understand much of this abstract, and may decide to not to read the paper (which would be too bad, since the paper is outstanding). As much as possible, these abbreviations should be defined and/or eliminated in favor of using the full word(s).

We have attempted to improve this while staying within the word limit confines.

7) Throughout the manuscript, the authors need to base their conclusions and their statements firmly on their empirical data. The present text should be extensively revised to eliminate overstatements and definitive conclusions. The authors have obtained nice data that in sum are consistent with their proposed structure, but they have presented the data in many places as definitive, and even included linkages for which no data are provided. The text should make the tentative nature of their conclusions abundantly clear. Final proof will only come when the transfer of ribitol, xylose, GlcA and matriglycan can be demonstrated in vitro, using purified enzymes, and fully characterized substrates and products. Until then, statements such as "Here we elucidate the structure" in the Abstract and "Here we define a fully functional glycan structure" are unacceptable. There remain many possibilities for the roles of FKTN and FKRP as well as other mysteries to explain, including why MEB disease arises from mutation of POMGNT1, which is apparently not part of the M3 core glycan.

We thank the reviewers for raising this important point and have significantly altered the text and figures to reflect the findings of our empirical data. Further, careful analysis of our data has revealed that HF can generate glycopeptides with 2 phosphates and thus we have not assigned the position of the ribitol-Xyl-GlcA-(Xyl-GlcA)_n_ to the phosphotrisaccharide though the recent publication by Toda and colleagues puts the structure in a phosphodiester linkage to the GalNAc residue.

8) The authors should be sure to report numbers of experimental replicates and n values for all of their figures; some (e.g. Figure 6) show single examples, and so it is important to indicate how many times such outcomes were observed.

We have added n values and defined error bars where appropriate.